# IMPROVED ALGORITHMS FOR REPLICABLE BANDITS

## ABSTRACT

This work is motivated by the growing demand for reproducible machine learning. We study the stochastic multi-armed bandit problem, where the algorithm's sequence of actions is, with a high probability, not affected by the randomness of the dataset. Existing algorithms require a regret scale of $O(K^3)$, which increases much faster than the number of actions (or "arms"), denoted as $K$. We introduce an algorithm with a distribution-dependent regret of $O(K)$ when the suboptimality gaps for each arm are within a constant factor. Furthermore, we propose another algorithm, which not only achieves a regret of $O(K)$ but also boasts a distribution-independent regret of $O(K^{1.5}\sqrt{T \log T})$. Additionally, we propose an algorithm for the linear bandit with regret of $O(d)$, which is linear in the dimension of associated features, denoted as $d$, and it is independent of $K$. For the analysis of these algorithms, we offer a principled approach to limiting the probability of non-replication, which clarifies the steps that existing research has implicitly followed.

## 1 INTRODUCTION

The *multi-armed bandit (MAB)* problem is one of the most well-known instances of sequential decision-making problems in uncertain environments, which can model various real-world scenarios. The problem involves conceptual entities called *arms*, of which there are a total of $K$. At each round $t = 1, 2, \dots$, the forecaster selects one of the $K$ arms and receives a corresponding reward. The forecaster's objective is to maximize the cumulative reward over these rounds. Maximizing this cumulative reward is equivalent to minimizing *regret*, the difference between the forecaster's cumulative reward and the reward of the best arm. The initial investigation of this problem took place within the field of statistics (Thompson, 1933; Robbins, 1952). In the past two decades, the machine learning community has conducted extensive research in this area, driven by numerous applications, including website optimization, A/B testing, and the formulation of meta-algorithms for algorithmic procedures (Auer et al., 2002; Li et al., 2010; Komiyama et al., 2015; Li et al., 2017).

Several algorithms have proven to be effective. Notably, the upper confidence bound (UCB, Lai & Robbins, 1985; Auer et al., 2002) and Thompson sampling (TS, Thompson, 1933) are widely recognized. Research has shown that these algorithms are asymptotically optimal (Cappé et al., 2013; Agrawal & Goyal, 2012; Kaufmann et al., 2012) in terms of their regret, meaning that these efficient algorithms exploit accumulated reward information to the fullest extent possible.

### 1.1 REPLICABILITY

One possible drawback of such efficiency is the algorithm's stability when dealing with small changes in the dataset, which can make replicating results challenging. To illustrate this, consider the following example:

**Example 1.** (Crowdsourcing (Abraham et al., 2013; Tran-Thanh et al., 2014)) Imagine a company conducting a crowd-based A/B testing with $K$ items. In this scenario, each round $t$ corresponds to a worker visiting their website, and each reward represents the feedback provided by the worker, such as a five-star rating. The key statistic of interest here is the mean score given by the workers. By using UCB or TS to allocate an item to each user, the system can quickly eliminate unpopular items from the candidate set. Although the company aims to publish the results, it is reluctant to disclose all the specific details of the setup. Therefore, it provides the experimental protocol along with summary statistics.

Table 1: Comparison of regret bounds in the $K$-armed and linear bandit problems. Means $\{\mu_i\}_{i \in [K]}$ are sorted in descending order, $\Delta_i = \mu_1 - \mu_i$ is the suboptimality gap of the arm $i$, and $\Delta = \Delta_2$. Algorithm 2 is referenced multiple times, implying that it has the smallest regret among the bounds. The lower bound is derived for the two-armed case. $\tilde{O}$ omits a polylog factor in $d, K, T$.

| Problem | Esfandiari et al. (2023a) | This work |
|---------|---------------------------|-----------|
| $K$-armed | $O\left(\sum_{i=2}^{K} \frac{1}{\Delta_i} \frac{K^2 \log T}{\rho^2}\right)$ $(= O(K^3))$ | $O\left(\sum_{i=2}^{K} \frac{\Delta_i}{\Delta^2} \frac{\log T}{\rho^2}\right)$ $(= O(K),$ REC (Theorem 3) and RSE (Theorem 4)) $O\left(\sum_{i=2}^{K} \frac{1}{\Delta_i} \frac{K^2 \log T}{\rho^2}\right)$ (RSE, Theorem 4) $O\left(\frac{K}{\rho}\sqrt{KT \log T}\right)$ (RSE, Theorem 4) $\Omega\left(\left(\frac{1}{\Delta}\right) \max\left(\log T, \frac{1}{\rho^2 \log((\rho\Delta)^{-1})}\right)\right)$ (Lower bound, Theorem 5) |
| Linear | $\tilde{O}\left(\frac{K^2\sqrt{dT}}{\rho^2}\right)$ $(= O(K^2))$ | $O\left(\frac{d \log T}{\Delta^2 \rho^2}\right)$ (RLSE, Theorem 8) $O\left(\frac{K}{\rho}\sqrt{dKT \log T}\right)$ (RLSE, Theorem 8) |

In Example 1, the original dataset is not disclosed, making it impossible for an external institution to perfectly replicate the experiment. To address this lack of guarantee due to the vague dataset specification, we focus on the algorithm's *stability*. Broadly speaking, a stable algorithm is robust against minor changes in the dataset. Notable categories of stability encompass differential privacy (Dwork et al., 2014), worst-case and average sensitivity (Varma & Yoshida, 2021), and pseudo-determinicity (Gat & Goldwasser, 2011). Among these notions of stability, we consider the *replicability* (Impagliazzo et al., 2022) in this work. In simple terms, a replicable algorithm exhibits almost identical behavior on two datasets sharing the same data-generating process. This concept aligns well with Example 1, where the data-generating process is clearly defined as a material method, whereas the dataset itself remains undisclosed.

Another advantage of promoting replicability in the context of sequential learning is its relation to statistical testing. It is well-known that the standard frequentist confidence interval no longer holds for the results of the multi-armed bandit problem because such an adaptive algorithm violates the assumption of statistical testing that the number of samples is fixed. In general, mean statistics derived from the multi-armed bandit algorithm are downward biased (Xu et al., 2013; Shin et al., 2019), and this bias persists even for a large sample scheme (Lai & Wei, 1982), rendering the use of standard confidence intervals ineffective even for asymptotics (Deshpande et al., 2018). Replicability is one of the best methods to address such bias because it forces the algorithm to exhibit identical behavior in multiple runs with different datasets sharing a common underlying data-generating process.

## 1.2 Our Contributions

The concept of replicability in learning was formalized by Impagliazzo et al. (2022). As far as we know, Esfandiari et al. (2023a) is the sole work that provides algorithms studying replicability for the multi-armed bandit problem. Roughly speaking, the regret of the algorithms therein is $O(K^2/\rho^2)$ times larger than that of non-replicable algorithms, which have $O(K)$ regret, such as UCB and TS. Here, $K$ is the number of arms, and $\rho$ is the probability of non-replication. Although the additional factor might appear necessary for the cost of replicability, it is somewhat disappointing because an $O(K^3)$-regret algorithm does not scale well with a moderate value of $K$. Upon closer examination of the problem, we discovered that $K^2$ factor can be eliminated, while the $1/\rho^2$ factor remains essential. The algorithmic contributions of this work are outlined as follows.

- We first introduce the general framework for bounding the probability of nonreplication.
- We introduce the Replicable Explore-then-Commit (REC) algorithm, which is the first replicable $K$-armed bandit algorithm with a regret bound of $O(K)$ when the suboptimality gaps for each arm are within a constant factor.

- While REC has an $O(K)$ bound, its distribution-dependent multiplicative factor may exceed that of existing algorithms in certain cases. To deal with this issue, we introduce the Replicable Successive Elimination (RSE) algorithm whose regret bound is the minimum of those of REC and the existing algorithms. Furthermore, we establish the distribution-independent regret bound for the replicable $K$-armed bandit problem.
- We derive the first lower bound for the replicable $K$-armed bandit problem, implying the necessity of the $1/\rho^2$ factor.
- Furthermore, we consider the linear bandit problem, in which each of the $K$ arms is associated with $d < K$ features. We show that a straightforward modification of RSE yields an algorithm with a regret bound of $O(d)$, indepenent of $K$.

A comparison of existing algorithms and our algorithms is summarized in Table 1.

## 2 PROBLEM SETUP

We consider the finite-armed stochastic bandit problem with $T$ rounds. At each round $t$, the forecaster who adopts an algorithm selects one of the arms $I_t \in [K] := \{1, 2, 3, \ldots, K\}$ and receives the corresponding reward $r_t$. Each arm $i \in [K]$ has an (unknown) mean parameter $\mu_i \in \mathbb{R}$. Here, $\mu_i \in [a, b]$ for some $a, b \in \mathbb{R}$ and let $S = b - a$. For ease of discussion, we assume $S = 1$, but all our results can be easily generalized[1] to any $S > 0$. The reward at round $t$ is $r_t = \mu_{I_t} + \eta_t$, where $\eta_t$ is a $\sigma$-subgaussian random variable that is independently drawn at each round.[2] The subgaussian assumption is quite general that is not limited to Gaussian random variables. Any bounded random variable is subgaussian, and thus, it is capable of representing binary events (Yes/No) and ordered choice (e.g., 5-star rating). For subgaussian random variables, the following inequality holds.

**Lemma 1** (Concentration inequality). *Let $X_1, X_2, \ldots, X_N$ be $N$ independent (zero-mean) $\sigma$-subgaussian random variables, and $\hat{\mu}_N = (1/N) \sum_i X_i$ be the empirical mean. Then, we have*

$$\mathbb{P}\left[|\hat{\mu}_N| \geq s\right] \leq 2 \exp\left(\frac{-s^2 N}{2\sigma^2}\right). \tag{1}$$

For ease of discussion, we assume the mean reward of each arm is distinct. In this case, we can assume $\mu_1 > \mu_2 > \cdots > \mu_K$ without loss of generality. Of course, an algorithm cannot exploit this ordering. A quantity called regret is defined as follows:

$$\text{Regret}(T) := \sum_{t \in [T]} (\mu_1 - \mu_{I_t}) = \sum_{i \in [K]} \Delta_i N_i(T),$$

where $\Delta_i = \mu_1 - \mu_i$ and $N_i(T)$ is the number of draws on arm $i$ during the $T$ rounds. We also denote $\Delta = \min_{i \geq 2} \Delta_i = \Delta_2$. The performance of an algorithm is measured by the expected regret, where the expectation is taken over (hypothetical) multiple runs. Before discussing the replicability, we formalize the notion of dataset in a sequential learning problem because the reward $r_t$ in the aforementioned procedure is drawn adaptively upon the choice of the arm $I_t$. The fact that each noise term $\eta_t$ is drawn independently enables us to reformulate the problem as follows:

**Definition 1.** (Dataset) The process of the multi-armed bandit problem is equivalent to the following: First, draw a matrix $(r_{i,n})_{i \in [K], n \in [T]}$, where $r_{i,n} = \mu_i + \eta_{i,n}$ and $\eta_{i,n}$ is a $\sigma$-subgaussian random variable. Second, Run a multi-armed bandit problem. Here, $r_t$ is the $(I_t, N_{I_t}(t))$-entry of the matrix. We call this matrix a *dataset* and denote it as $\mathcal{D}$. We call $(\mu_i)_{i \in [K]}$ a data-generating process or a model.

Following Esfandiari et al. (2023a), we consider the class of replicable algorithms that, with high probability, gives exactly the same sequence of selected arms for two independent runs.

**Definition 2.** ($\rho$-replicability, Impagliazzo et al. (2022); Esfandiari et al. (2023a)) For $\rho \in [0, 1]$, an algorithm is $\rho$-replicable if,

$$\mathbb{P}_{U, \mathcal{D}^{(1)}, \mathcal{D}^{(2)}}\left[(I_1^{(1)}, I_2^{(1)}, \ldots, I_T^{(1)}) = (I_1^{(2)}, I_2^{(2)}, \ldots, I_T^{(2)})\right] \geq 1 - \rho, \tag{2}$$

---

[1]In particular, $\epsilon_p$ of Algorithms 1–3 should be replaced with $\epsilon_p = S2^{-p}$ and all the other parts remain the same.

[2]A random variable $\eta$ is $\sigma$-subgaussian if $\mathbb{E}[\exp(\lambda\eta)] \leq \exp(\sigma^2\lambda^2/2)$ for any $\lambda$. For example, a zero-mean Gaussian random variable with variance $\sigma^2$ is $\sigma$-subgaussian.

where $U$ represents the internal randomness, and $\mathcal{D}^{(1)}, \mathcal{D}^{(2)}$ are the two datasets that are drawn from the same data-generating process $\{\mu_i\}_{i \in [K]}$.

Here, we may consider $U$ as a sequence of uniform random variables on $[0, 1]$ that the algorithm can use to control its behavior. For an algorithm to be replicable, the use of such random variables is crucial. The value $\rho$ corresponds to the probability of misreplication. The smaller $\rho$ is, the more likely the sequence of actions is replicated. By definition, any algorithm is 1-replicable, and no nontrivial algorithm is 0-replicable.[3] In this paper, we consider $\rho \in (0, 1)$ as an exogenous parameter, and our goal is to minimize the regret subject to the $\rho$-replicability.

# 3    General Bound of the Probability of Non-replication $\rho$

It is not very difficult to see that a standard bandit algorithm, such as UCB, lacks replicability. UCB, in each round, compares the UCB index of the arms, and thus, a minor change in the dataset can alter the sequence of draws $I_1, I_2, \dots, I_T$. Thus, designing a replicable algorithm must deviate significantly from standard bandit algorithms. This section presents a general framework for bounding non-replicable probability in the multi-armed bandit problem. We believe that this framework can be applied to many other sequential learning problems. First, a replicable algorithm should limit its flexibility by introducing phases.

**Definition 3.** A set of phases is a consecutive partition of rounds $[T]$. Namely, phase $p$ is a consecutive subset of $[T]$, and the first round of phase $p + 1$ follows the last round of phase $p$, and each round belongs to one of the phases. We define $P$ to be the number of phases.

The sequence of draws $I_1, I_2, \dots, I_T$ is only allowed branch at the end of each phase, which we formalize in the following definition.

**Definition 4.** (Randomness) The randomness $U$ consists of the one for each phase. Namely, $U = (U_1, U_2, \dots, U_P)$.

**Definition 5.** (Good events, decision variables, and decision points) We call the end of the final round of each phase a decision point, which we denote as $T_p$. For each $p \in [P]$, we consider the history $\mathcal{H}_p$ to be the set of all results up to the final round $T_p$ of phase $p$. Namely,

$$\mathcal{H}_p = (I_1, r_1, I_2, r_2, \dots, I_{T_p}, r_{T_p}) \cup (U_1, U_2, \dots, U_p). \tag{3}$$

Each phase $p$ is associated with good event $\mathcal{G}_p(\mathcal{H}_p)$, which is a binary function of $\mathcal{H}_p$. Each phase $p$ is associated with random variables that are called *decision variables* $d_p$. Decision variables take discrete values and are functions of $\mathcal{H}_p$. Moreover, the sequence of draws on the next phase $\{I_{T_p+1}, I_{T_p+2}, \dots, I_{T_{p+1}}\}$ is uniquely determined by the decision variables $d_1, d_2, \dots, d_p$.

Intuitively speaking, the good events correspond to the concentration of statistics with its probability we can bound with concentration inequalities (by Lemma 1). The set of decision variables uniquely determines the sequence of draws. Note that each phase can be associated with more than one decision variable. To obtain intuition, we consider the following example.

**Example 2.** (A replicable elimination algorithm, Alg 2. in Esfandiari et al. (2023a)) At the end of each phase, the algorithm obtains an empirical estimate of $\mu_i$ for each arm. It tries to eliminate suboptimal arm $i$, and the corresponding decision variable is

$$d_{p,i} = \mathbf{1}[\max_j \mathrm{LCB}_j(p) \geq \mathrm{UCB}_i(p)], \tag{4}$$

where $\mathrm{UCB}_i(p), \mathrm{LCB}_i(p)$ are the (randomized) upper/lower confidence bounds of the arm $i$ at phase $p$. Here, $\mathbf{1}[\mathcal{E}]$ is 1 if event $\mathcal{E}$ holds or 0 otherwise. Under good events, by randomizing the confidence bounds with $U_p$, it bounds the probability of non-replication of each decision variable.

In the following, we defined the non-replication probability for each component.

**Definition 6.** (Probability of bad event) Let $\rho_p^G = \mathbb{P}\left[\mathcal{G}_p^c\right]$, where $\mathcal{G}^c$ is a complement event of $\mathcal{G}$.

---

[3]A 0-replicable algorithm draws an identical sequence of arms for almost all fixed $U$.

**Definition 7.** (Non-replication probability of a decision variable) Let $d_{p,i}$ be the $i$-th decision variable at phase $p$. Its non-replication probability $\rho_{p,i}$ is defined as

$$\rho^{(p,i)} := \mathbb{P}_{U, \mathcal{D}^{(1)}, \mathcal{D}^{(2)}} \left[ d_p^{(1)} \neq d_p^{(2)} \; \middle| \; \bigcap_{p'=1}^{p-1} \left\{ d_{p'}^{(1)} = d_{p'}^{(2)}, \mathcal{G}_{p'}^{(1)}, \mathcal{G}_{p'}^{(2)} \right\}, \mathcal{G}_p^{(1)}, \mathcal{G}_p^{(2)}, \right] \quad (5)$$

where we use superscripts $(1)$ and $(2)$ for the corresponding variables on the two datasets $\mathcal{D}^{(1)}, \mathcal{D}^{(2)}$.

**Theorem 2.** (Replicability of an algorithm) *An algorithm is $\rho$-replicabile with*

$$\rho \leq 2 \sum_p \rho_p^G + \sum_{p,i} \rho^{(p,i)}. \quad (6)$$

In summary, Theorem 2 enables us to decompose the non-replication probability $\rho$ into the sum of the non-replication probabilities due to the bad events ($\mathcal{G}^c$) and decision variables ($\rho^{(p,i)}$).

### 3.1 COMPARISION OF ALGORITHMS IN VIEW OF DECISION VARIABLES

The smaller the non-replication probability of each decision variable is, the higher the cost the algorithm must pay to guarantee it. Assuming all $\rho^{(p,i)}$ are equal, $\rho^{(p,i)} \sim \rho/|\{\rho^{(p,i)}\}|$, where $|\{\rho^{(p,i)}\}|$ is the number of decision variables. Algorithm 2 in Karbasi et al. (2023) uses the decision variables for eliminating each suboptimal arm. Therefore, it has $\tilde{O}(K)$ decision variables, implying that each of the non-replication probability must be $\tilde{O}(\rho/K)$. As a result, these algorithms has an $O(K^2/\rho^2)$ factor in the leading term of the regret. In our Replicable Explore-the-Commit (REC) algorithm, we use decision variables representing whether or not to finish the exploration process, which means that we only need $O(1)$ (in fact, only one of them is effective!) decision variables with its non-replication level $O(\rho)$, and as a result, it has $O(1/\rho^2)$ factor in the leading term of the regret, which dramatically reduces the dependence on $K$.

## 4 AN $O(K)$-REGRET ALGORITHM FOR $K$-ARMED BANDIT PROBLEM

This section introduces the Replicable Explore-then-Commit (REC, Algorithm 1), an $O(K)$-regret algorithm for the $K$-armed bandit problem. This algorithm consists of multiple exploration phases and an exploitation period. The last round of each phase is a decision point, where the algorithm decides whether it terminates the exploration period or not. For this aim, it utilizes the minimum suboptimality gap estimator $\hat{\Delta}(p) = \max_i \hat{\mu}_i(p) - \max_i^{(2)} \hat{\mu}_i(p)$, where $\max_i^{(2)}$ denotes the second largest element. This algorithm involves a single uniform random variable $U_p \sim \text{Unif}(0,1)$ for each phase $p$.

Let $\text{Conf}(p) := \epsilon_p/C_\rho$ and $\epsilon_p = 2^{-p}$. Here, the universal constant $C_\rho$ is clarified later in Theorem 3. At phase $p$, if the algorithm is in the exploration period, we draw each arm up to

$$N_p := 4\sigma^2 \frac{\log(KPT)}{(\text{Conf}(p))^2} = O\left(4^p \log T\right) \quad (7)$$

times, where $P = \min_p\{N_p \geq T\} = O(\log T)$ is the maximum number of phases. Lemma 1 with $s = \text{Conf}(p)$ implies that, with probability at least $1 - 1/PT$, we have $|\hat{\Delta}_{ij}(p) - \Delta_{ij}| \leq \text{Conf}(p)$, for each gap $\Delta_{ij} = |\mu_i - \mu_j|$ and its empirical estimator $\hat{\Delta}_{ij}(p) = |\hat{\mu}_i(p) - \hat{\mu}_j(p)|$.

---

**Algorithm 1:** Replicable Explore-then-Commit (REC)

---

```
// Exploration period
```
1 **for** $p = 1, 2, \ldots, P$ **do**
2      $\epsilon_p = 2^{-p}$ ;
3      Draw shared random variable $U_p \sim \text{Unif}(0,1)$;
4      Draw each arm for $N_p$ times;
5      If $\hat{\Delta}(p) \geq \frac{(2+U_p)\epsilon_p}{\rho}$, then fix the estimated best arm $\hat{i}^* \in \arg\max_i \hat{\mu}_i(p)$ and break the
     loop. Note that $\hat{\Delta}(p) > 0$ implies $|\mathcal{A}_{p+1}| = 1$.
```
// Exploitation period
```
6 Draw arm $\hat{i}^*$ for the rest of the rounds.

---

The following theorem guarantees the replicability and performance of Algorithm 1.

**Theorem 3.** *Let $C_\rho \geq 9/4$. Assume that $\rho \leq 1/2$ and $T \geq 36K/\rho$. Then, Algorithm 1 is $\rho$-replicable and the following regret bound holds:*

$$\mathbb{E}[\text{Regret}(T)] = O\left(\sum_{i=2}^{K} \frac{\Delta_i}{\Delta^2} \frac{\log T}{\rho^2}\right). \tag{8}$$

**Remark 1.** (Use of explore-then-commit) Esfandiari et al. (2023a) briefly remarked the possibility of the use of the explore-then-commit strategy and sketched an $O(T \sum_i \Delta_i)$ regret, from which we significantly improved the regret as well as getting rid of the assumption of known $\Delta$.

## 5    A GENERALIZED ALGORITHM FOR $K$-ARMED BANDIT PROBLEM

---
**Algorithm 2:** Replicable Successive Elimination (RSE)

---
1  Initialize the candidate set $\mathcal{A}_1 = [K]$;
2  **for** $p = 1, 2, \ldots, P$ **do**
3  $\quad$ $\epsilon_p = 2^{-p}$;
4  $\quad$ Draw shared random variables $U_{p,i} \sim \text{Unif}(0,1)$ for $i = 0, 1, 2, \ldots, K$ ;
5  $\quad$ Draw each arm in $\mathcal{A}_p$ up to $N_p$ times;
6  $\quad$ $\mathcal{A}_{p+1} \leftarrow \mathcal{A}_p$;
7  $\quad$ **if** $\hat{\Delta}(p) \geq \frac{(2+U_{p,0})\epsilon_p}{\rho_a}$ **then**
8  $\quad\quad$ $\mathcal{A}_{p+1} = \{\arg\max_i \hat{\mu}_i(p)\}$;$\quad\quad\quad$ ▷ Eliminate all arms except for one. By definition of
$\quad\quad$ $\hat{\Delta}(p) > 0, |\mathcal{A}_{p+1}| = 1$ holds.
9  $\quad$ **for** $i \in \mathcal{A}_{p+1}$ **do**
10 $\quad\quad$ **if** $\hat{\Delta}_i(p) \geq \frac{(2+U_{p,i})\epsilon_p}{\rho_e}$ **then**
11 $\quad\quad\quad$ $\mathcal{A}_{p+1} \leftarrow \mathcal{A}_{p+1} \setminus \{i\}$;$\quad\quad\quad\quad\quad$ ▷ Eliminate arm $i$.

---

Although Algorithm 1 improves the existing $O(K^3)$ regret bound in terms of the dependency to $K$, there are two potential drawbacks: First, when we further compare the ratio between REC's regret bound with the existing regret bound of

$$O\left(\sum_{i=2}^{K} \frac{1}{\Delta_i} \frac{K^2 \log T}{\rho^2}\right) \quad \text{(Alg.2 in Esfandiari et al. (2023a))}, \tag{9}$$

the ratio between the two distribution-dependent bounds is bounded as (equation 8/equation 9) $\leq$ $((\Delta_K)/(K\Delta))^2$, which implies that REC may be inferior if the minimum suboptimality gap $\Delta$ is extremely small (i.e., $\Delta \ll (\Delta_K)/K$). Second, it does not have a distribution-independent regret bound. To address these issues, we introduce Algorithm 2, which generalizes Algorithm 1. Unlike Algorithm 1, it keeps the list of remaining arms $\mathcal{A}_p$ that it draws. At the end of each phase, it attempts to eliminate all but one arm (Line 7). If that fails, it attempts to eliminate each arm $i$ (Line 10). Here, $\hat{\Delta}_i(p) = \max_j \hat{\mu}_j(p) - \hat{\mu}_i(p)$ be the estimated suboptimality gap. Here, the hyperparameters $\rho_a, \rho_e$ therein determine the confidence level for elimination. One can confirm that Algorithm 1 is a specialized version of Algorithm 2 where $(\rho_a, \rho_e) = (\rho, 0)$, where $\rho_e = 0$ implies the corresponding elimination never occurs. Here, eliminating all but one arm is equivalent to switching to the exploration period. However, when $\rho_e > 0$, it attempts to eliminate each arm as well. The following theorem guarantees the replicability and the regret of Algorithm 2.

**Theorem 4.** *Let $C_\rho \geq 9/4$, $\rho_a = \rho/2$, and $\rho_e = \rho/(2 \max(K-2, 2))$. Assume that $\rho \leq 1/2$ and $T \geq 36K/\rho$. Then, Algorithm 2 is $\rho$-replicable. Moreover, the following three regret bounds hold:*

$$\mathbb{E}[\text{Regret}(T)] = O\left(\sum_{i=2}^{K} \frac{\Delta_i}{\Delta^2} \frac{\log T}{\rho^2}\right), \quad \text{(same as equation 8)} \tag{10}$$

$$\mathbb{E}[\text{Regret}(T)] = O\left(\sum_{i=2}^{K} \frac{K^2 \log T}{\Delta_i \rho^2}\right), \quad \text{(same as equation 9)} \tag{11}$$

$$\mathbb{E}[\text{Regret}(T)] = O\left(\frac{K}{\rho}\sqrt{KT\log T}\right). \quad \textit{(distribution-independent regret)} \tag{12}$$

In other words, Algorithm 2 has the best bound of Algorithm 1 and existing algorithms. Moreover, this algorithm is the first replicable algorithm that has a distribution-independent regret bound in the $K$-armed bandit problem.

## 6   REGRET LOWER BOUND FOR REPLICABLE ALGORITHMS

This section provides the regret lower bound for $K$-armed bandit algorithms. Following the literature, we consider the class of uniformly good algorithms. Intuitively speaking, an algorithm is uniformly good if it works with any model $(\mu_1, \mu_2, \ldots, \mu_K)$.

**Definition 8** (Uniformly good, Lai & Robbins (1985))**.** An algorithm is *uniformly good*, if for any $a > 0$ and for any model $(\mu_1, \mu_2, \ldots, \mu_K)$, there exists a function $R(T) = o(T^a)$ such that

$$\mathbb{E}[\text{Regret}(T)] \leq R(T). \tag{13}$$

**Theorem 5.** *Consider a two-armed bandit problem where reward is drawn from* $\text{Bernoulli}(\mu_i)$ *for each arm* $i = 1, 2$ *with mean parameters* $\mu_1, \mu_2$. *Consider an algorithm that is uniformly good and* $\rho$-*replicable. Then, for any* $\Delta > 0$, *there exists an instance* $(\mu_1, \mu_2)$ *with* $\Delta = |\mu_1 - \mu_2|$ *such that the regret of any* $\rho$-*replicable bandit algorithm is lower-bounded as*

$$\mathbb{E}[\text{Regret}(T)] = \Omega\left(\frac{1}{\rho^2\Delta\log((\rho\Delta)^{-1})}\right). \tag{14}$$

This bound implies that REC, RSE, and the algorithms in Esfandiari et al. (2023a) are optimal up to a polylogarithmic factor for two-armed bandit problem.

**Remark 2.** It is well-known that another lower bound

$$\mathbb{E}[\text{Regret}(T)] = \Omega\left(\frac{\log T}{\Delta}\right) \tag{15}$$

holds for a uniformly good algorithm (c.f., Theorem 1 in Lai & Robbins (1985)). Therefore, a lower bound for a $\rho$-replicable uniformly good algorithm is the maximum of equation 14 and equation 15.

The absence of $\log T$ in equation 14 appears to be essential. The factor $\log T$ is derived from the uniformly good property of a bandit algorithm. However, the cost of replicability and the cost of uniform goodness are not necessarily compounded. Any $\rho$-replicable algorithm (if it is not uniformly good) that frequently selects the best arm should maintain the lower bound of equation 14.

## 7   AN ALGORITHM FOR LINEAR BANDIT PROBLEM

Next, we consider the linear bandit problem, a special version of the $K$-armed bandit problem where associated information is available. In this problem, each arm $i \in [K]$ is associated with a $d$-dimensional feature vector $\boldsymbol{x}_i \in \mathbb{R}^d$ and the reward $r_t$ of choosing an arm $I_t$ is $\boldsymbol{x}_{I_t}^\top\boldsymbol{\theta} + \eta_t$, where $\boldsymbol{\theta}$ is (unknown) shared parameter vector, and $\eta_t$ is a $\sigma$-subgaussian random variable. Namely, the mean $\mu_i = \boldsymbol{x}_i^\top\boldsymbol{\theta}$ can be estimated via known feature $\boldsymbol{x}_i$ and unknown shared coefficients $\boldsymbol{\theta}$. Without loss of generality, we assume $\text{span}(\{x_i\}_{i=1}^K) = \mathbb{R}^d$.

We introduce the replicable linear successive elimination (RLSE). Similarly to RSE (Algorithm 2), this algorithm is elimination-based. The main innovation here is to use the G-optimal design that explores all dimensions in an efficient way. Namely,

**Definition 9.** (G-optimal design) For $\mathcal{A}_p \subseteq [K]$, let $\pi$ be a distribution over $\mathcal{A}_p$. Let

$$\boldsymbol{V}(\pi) = \sum_{i \in \mathcal{A}_p} \pi(\boldsymbol{x}_i)\boldsymbol{x}_i\boldsymbol{x}_i^\top, \quad g(\pi) = \max_{i \in \mathcal{A}_p} ||\boldsymbol{x}_i||_{\boldsymbol{V}(\pi)^{-1}}^2. \tag{16}$$

A distribution $\pi^*$ is called a G-optimal design if it minimizes $g$, i.e., $\pi^* \in \arg\min_\pi g(\pi)$.

We use the following well-known results for G-optimal designs (See, e.g., Section 21 of Lattimore & Szepesvári (2020)).

**Lemma 6** (Kiefer-Wolfowitz). *A G-optimal design $\pi^*$ satisfies $g(\pi^*) = d$.*

In this paper, we assume the availability of a constant approximation of optimal design $\pi^{*,\mathrm{app}} = \pi^{*,\mathrm{app}}(\mathcal{A}_p)$ with $g(\pi^{*,\mathrm{app}}) \leq 2d$. An explicit construction of such an approximated G-optimal design is found in the literature (e.g., Lemma 7 of (Esfandiari et al., 2023a)). Given an oracle for an approximated G-optimal design, we define the allocation at phase $p$ to be $N_i^{\mathrm{lin}}(p) = \lceil N^{\mathrm{lin}}(p)\pi_i^{*,\mathrm{app}} \rceil$ ,where

$$N^{\mathrm{lin}}(p) := \frac{16\sigma^2 d \log(|\mathcal{A}_p|PT)}{(\mathrm{Conf}(p))^2}. \tag{17}$$

Note that $\sum_i N_i^{\mathrm{lin}}(p) \leq N^{\mathrm{lin}}(p) + K$. We use the following lemma for the confidence bound (see e.g., Section 21.1 of Lattimore & Szepesvári (2020)):

**Lemma 7** (Fixed-sample bound). *Consider the estimator $\hat{\boldsymbol{\theta}}_p$ at the end of phase $p$. Then, with probability at least $1 - \frac{2}{PT}$, the following bound holds uniformly for any $i \in \mathcal{A}_p$:*

$$|\boldsymbol{x}_i^\top(\boldsymbol{\theta} - \hat{\boldsymbol{\theta}}_p)| \leq \frac{\mathrm{Conf}(p)}{2}. \tag{18}$$

Letting $\hat{\mu}_i = \boldsymbol{x}_i^\top \hat{\boldsymbol{\theta}}_p$ and $\hat{\Delta}_{ij} = |\hat{\mu}_i - \hat{\mu}_j|$, equation 18 implies

$$|\Delta_{ij} - \hat{\Delta}_{ij}| \leq \mathrm{Conf}(p). \tag{19}$$

Apart from applying approximated G-optimal exploration, the algorithm closely mirrors the steps of RSE. A comprehensive description of RLSE can be found in Appendix A. The subsequent theorem assures both the replicability and regret of RLSE.

**Theorem 8.** *Let $C_\rho \geq 9/4$, $\rho_a = \rho/2$, and $\rho_e = \rho/(2\max(K-2,2))$. Assume that $\rho \leq 1/2$ and $T \geq 36K/\rho$. Then, RLSE is $\rho$-replicable. Moreover, the following two regret bounds hold:*

$$\mathbb{E}[\mathrm{Regret}(T)] = O\left(\frac{d\log T}{\Delta^2\rho^2}\right), \quad \textit{(an $O(d)$ distribution-dependent bound)} \tag{20}$$

$$\mathbb{E}[\mathrm{Regret}(T)] = O\left(\frac{K}{\rho}\sqrt{dKT\log T}\right). \quad \textit{(distribution-independent bound)} \tag{21}$$

The first bound depends on the suboptimality gap $\Delta$ and is independent of $K$. The second bound is distribution-independent and is smaller than the existing bound by a $\sqrt{K}/\rho$ factor (c.f., Table 1).

## 8 SIMULATION

We compared our REC (Algorithm 1) and RSE (Algorithm 2) with RASMAB (Algorithm 2 of Esfandiari et al. (2023a), "Replicable Algorithm for Stochastic Multi-Armed Bandits"). We did not include Algorithm 1 of Esfandiari et al. (2023a) because its regret bound is always inferior to RASMAB. Three models of $K$-armed Gaussian bandit problems were considered. To ensure fair comparison, as RASMAB relies on the Hoeffding inequality, we standardized the variance of the arms at $0.5$. The results were averaged over 100 runs. The algorithms can be characterized as follows: REC eliminates all arms simultaneously, RASMAB eliminates each arm independently, and RSE incorporates both strategies. Theoretical results suggest that REC outperforms RASMAB provided that $\Delta_K/\Delta = o(K)$. We optimize $C_\rho$ in REC and RSE and $\beta$ of RASMAB for $\hat{\rho} = 0.3$ by using a grid search. Here, the empirical nonreplication probability $\hat{\rho}$ is obtained by bootstrapping. Namely, assume that the algorithm results in $S$ different sequences of draws, where the corresponding number of occurrences for each sequence are $N_{(1)}, N_{(2)}, N_{(3)}, \ldots, N_{(S)}$. By definition, $\sum_s N_{(s)} = 100$. Then, $\hat{\rho} := 1 - \sum_s (N_{(s)}/100)^2$.

We set the mean parameters as follows: $\boldsymbol{\mu} = (0.1, 0.1, 0.8, 0.8, 0.9)$ for Model 1, $\boldsymbol{\mu} = (0.1, 0.1, 0.5, 0.5, 0.9)$ for Model 2, and $\boldsymbol{\mu} = (0.0, 0.0, 0.0, 0.0, 0.0, 0.0, 0.0, 0.0, 0.0, 0.9)$ for Model 3. The amount of regret is depicted in Figure 1. A lower regret signifies superior performance. As

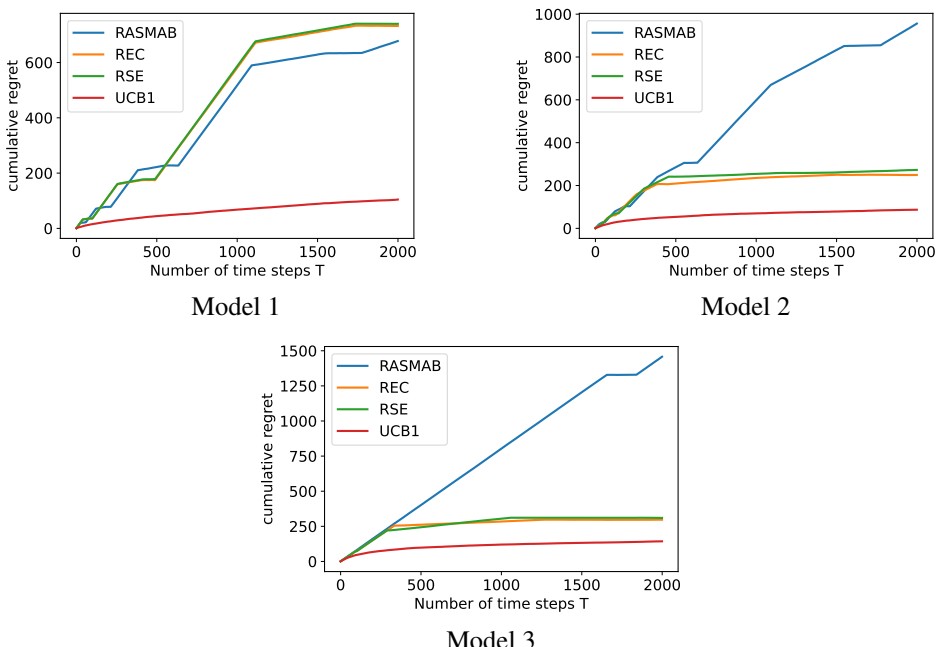

Figure 1: Regret of algorithms. The horizontal axis indicates the number of rounds $t$ from 1 to $T$, whereas the vertical axis indicates $\text{Regret}(t)$. Results of REC and RSE in Model 1 are very similar.

a non-replicable algorithm, UCB1 naturally outperforms all other replicable algorithms. Model 1, with a large $\Delta/\Delta_2 = 0.8/0.1 = 8$, is designed to favor RASMAB while Model 3, having $\Delta/\Delta_2 = 1$, is tailored to favor REC. In Models 2 and 3, REC and RSE significantly surpass RAS-MAB, indicating their success in replicably selecting the optimal arm. In Model 1, RASMAB marginally outperforms REC and RSE. As RASMAB is designed to eliminate each arm independently, the early elimination of clearly suboptimal arms (specifically, arms 1 and 2) decreases cumulative regret. These findings align with our theoretical results. Additional simulations employing theoretically-chosen hyperparameters are in the appendix.

## 9 RELATED WORK

Replicability was introduced by Impagliazzo et al. (2022) and they designed replicable algorithms for answering statistical queries, identifying heavy hitters, finding median, and learning halfspaces. Since then, replicable algorithms have been studied for bandit problems (Karbasi et al., 2023), reinforcement learning (Eaton et al., 2023), and clustering (Esfandiari et al., 2023b). The equivalence of various stability notions, including replicability and differential privacy (Dwork et al., 2014) was shown for a broad class of statistical problems (Bun et al., 2023). However, the equivalence therein does not necessarily guarantee an efficient conversion. Kalavasis et al. (2023) considered a relaxed notion of replicability. Note also that there are several relevant works Dixon et al. (2023); Chase et al. (2023) that study a different notion of applicability.

Prior to the introduction of the replicable bandit algorithm, the batched bandit problem was considered (Auer & Ortner, 2010; Cesa-Bianchi et al., 2013; Komiyama et al., 2013; Perchet et al., 2016; Gao et al., 2019; Esfandiari et al., 2021). In this problem, the algorithm needs to determine the sequence of draws at the beginning of each batch. Existing replicable bandit algorithms in Esfandiari et al. (2023a), as well as our algorithms, adopt phased approaches, and one can find similarities in the algorithmic design. In particular, Perchet et al. (2016) considered the two-armed batched bandit problem. They utilized the fact that the termination of the exploration phases in EtC only occurs in a fixed number of rounds, a concept that we also utilize in the proof of our algorithms. However, their algorithm does not guarantee $\rho$-replicability for $\rho < 1/2$. Our REtC extends their results by introducing a randomized confidence level to guarantee a further level of applicability. Furthermore, our RSE generalizes both EtC and successive elimination (Gao et al.,

2019; Esfandiari et al., 2023a) in a replicable way, and we recover essentially the same performance towards a large value of $\rho$.

Several different notions of stability have been explored in the context of sequential learning. For instance, robustness against corrupted distributions has been examined in the multi-armed bandit problem (Kim & Lim, 2016; Gajane et al., 2018; Kapoor et al., 2019; Basu et al., 2022). Differential privacy has also been considered in this context (Shariff & Sheffet, 2018; Basu et al., 2019; Hu & Hegde, 2022). Differential privacy considers the change of decision against the change of a single data point, whereas in the replicable bandits, we have more than one change of data points between two datasets that are generated from the identical data-generating process. Recent work (Dong & Yoshida, 2023) showed that an algorithm with a low average sensitivity (Varma & Yoshida, 2021) can be transformed to an online learning algorithm with low regret and inconsistency in the random-order setting, and hence in the stochastic setting.

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
