## A    REPLICABLE LINEAR SUCCESSIVE ELIMINATION

The Replicable Linear Successive Elimination (RLSE) algorithm is described in Algorithm 3.

---

**Algorithm 3:** Replicable Linear Successive Elimination (RLSE)

---

1  Initialize the candidate set $\mathcal{A}_1 = [K]$;
2  **while** $p = 1, 2, \ldots, P$ **do**
3       $\epsilon_p = 2^{-p}$;
4       Draw shared random variables $U_{p,i} \sim \mathrm{Unif}(0, 1)$ for each $i = 0, 1, 2, \ldots, K$;
5       Draw each arm for $N_i^{\mathrm{lin}}(p)$ times;           ▷ Approximated G-optimal exploration
6       $\mathcal{A}_{p+1} \leftarrow \mathcal{A}_p$;
7       **if** $\hat{\Delta}(p) \geq \frac{(2+U_{p,0})\epsilon_p}{\rho_a}$ **then**
8           $\mathcal{A}_{p+1} = \{\arg\max_i \hat{\mu}_i(p)\}$;     ▷ Eliminate all arms except for one. Note that $\hat{\Delta}(p) > 0$
            implies $|\mathcal{A}_{p+1}| = 1$.
9       **for** $i \in \mathcal{A}_{p+1}$ **do**
10          **if** $\hat{\Delta}_i(p) \geq \frac{(2+U_{p,i})\epsilon_p}{\rho_e}$ **then**
11             $\mathcal{A}_{p+1} \leftarrow \mathcal{A}_{p+1} \setminus \{i\}$;               ▷ Eliminate arm $i$.

---

## B    PROOFS ON GENERAL BOUND

*Proof of Theorem 2.* Let $\mathcal{G} = \bigcap_p \mathcal{G}_p$. We have

$$\rho := \mathbb{P}_{U,\mathcal{D}^{(1)},\mathcal{D}^{(2)}} \left[ (I_1^{(1)}, I_2^{(1)}, \ldots, I_T^{(1)}) \neq (I_1^{(2)}, I_2^{(2)}, \ldots, I_T^{(2)}) \right] \tag{22}$$

$$\leq \mathbb{P}[(I_1^{(1)}, I_2^{(1)}, \ldots, I_T^{(1)}) \neq (I_1^{(2)}, I_2^{(2)}, \ldots, I_T^{(2)}), \mathcal{G}^{(1)}, \mathcal{G}^{(2)}] + \mathbb{P}_{\mathcal{D}^{(1)}} \left[ \cup_p \mathcal{G}_p^c \right] + \mathbb{P}_{\mathcal{D}^{(2)}} \left[ \cup_p \mathcal{G}_p^c \right] \tag{23}$$

$$\leq \mathbb{P}[(I_1^{(1)}, I_2^{(1)}, \ldots, I_T^{(1)}) \neq (I_1^{(2)}, I_2^{(2)}, \ldots, I_T^{(2)}), \mathcal{G}^{(1)}, \mathcal{G}^{(2)}] + 2 \sum_p \rho_p^G \tag{24}$$

$$\text{(by Definition 6)} \tag{25}$$

$$\leq \mathbb{P} \left[ (d_p^{(1)})_{p=1}^P \neq (d_p^{(2)})_{p=1}^P, \mathcal{G}^{(1)}, \mathcal{G}^{(2)} \right] + 2 \sum_p \rho_p^G \tag{26}$$

$$\text{(by definition of decision variables)} \tag{27}$$

$$\leq \sum_p \mathbb{P} \left[ d_p^{(1)} \neq d_p^{(2)}, \cap_{p'=1}^{p-1} \left\{ d_{p'}^{(1)} = d_{p'}^{(2)} \right\}, \mathcal{G}^{(1)}, \mathcal{G}^{(2)} \right] + 2 \sum_p \rho_p^G \tag{28}$$

$$\leq \sum_p \mathbb{P} \left[ d_p^{(1)} \neq d_p^{(2)}, \cap_{p'=1}^{p-1} \left\{ d_{p'}^{(1)} = d_{p'}^{(2)}, \mathcal{G}_{p'}^{(1)}, \mathcal{G}_{p'}^{(2)} \right\}, \mathcal{G}_p^{(1)}, \mathcal{G}_p^{(2)} \right] + 2 \sum_p \rho_p^G \tag{29}$$

$$\leq \sum_p \mathbb{P} \left[ d_p^{(1)} \neq d_p^{(2)} \middle| \cap_{p'=1}^{p-1} \left\{ d_{p'}^{(1)} = d_{p'}^{(2)}, \mathcal{G}_{p'}^{(1)}, \mathcal{G}_{p'}^{(2)} \right\}, \mathcal{G}_p^{(1)}, \mathcal{G}_p^{(2)} \right] + 2 \sum_p \rho_p^G \tag{30}$$

$$\leq \sum_{p,i} \rho^{(p,i)} + 2 \sum_p \rho_p^G \tag{31}$$

$$\text{(by Definition 7)}. \tag{32}$$

$\square$

## C    PROOFS ON ALGORITHM 1

### C.1    REPLICABILITY OF ALGORITHM 1

This section bounds the probability of non-replication of Theorem 3. We use the general bound of Section 3. Let the good event (Definition 6) be

$$\mathcal{G} = \bigwedge_{p \in [P]} \mathcal{G}_p \tag{33}$$

$$\mathcal{G}_p = \bigwedge_{i,j \in [K]} \left\{ |\Delta_{ij} - \hat{\Delta}_{ij}(p)| \le \mathrm{Conf}(p) \right\}. \tag{34}$$

Event $\mathcal{G}$ states that all estimators lie in the confidence region.

**Lemma 9.** *Event $\mathcal{G}$ holds with probability at least $1 - \rho/18$.*

It is easy to see that the only decision variable at phase $p$ is

$$d_{(p,0)} := \mathbf{1} \left[ \hat{\Delta}(p) \ge \frac{(2 + U_p)\epsilon_p}{\rho} \right].$$

**Lemma 10.** *Let*

$$p_{\mathrm{s}} = \min_p \left\{ \epsilon_p \le \frac{10\Delta}{17\rho} \right\}.$$

*Under $\mathcal{G}$, the misreplication probability is $\rho^{(p,0)} = 0$ for any $p \ne p_{\mathrm{s}}$.*

Lemma 10 states that the only effective decision variable is that of phase $p_{\mathrm{s}}$.

**Lemma 11.** *Under $\mathcal{G}$, the probability of non-replication at each decision point is at most $\rho^{(p,0)} = 8\rho/9$ for $p = p_{\mathrm{s}}$.*

*Proof of non-replicability part of Theorem 3.* Theorem 2 and Lemmas 9–11 imply that the probability of misidentification is at most $2 \times \rho/18 + 8\rho/9 = \rho$, which completes the proof. $\qquad\square$

In the following, we derive Lemmas 9–11.

*Proof of Lemma 9.* Since $\Delta_{ij} = \mu_i - \mu_j$, $\hat{\Delta}_{ij} - \Delta_{ij}$ is estimating the sum of two $\sigma$-subgaussian random variables, which is a $2\sigma$-subgaussian random variable. By using Lemma 1 and taking a union bound over all possible $K(K-1)/2$ pairs of $ij$ and phases $1, \dots, P$, Event $\mathcal{G}$ holds with high probability:

$$\mathbb{P}[\mathcal{G}] \ge 1 - \frac{2K}{T}, \tag{35}$$

By assumption, $2K/T \le \rho/18$, which completes the proof. $\qquad\square$

*Proof of Lemma 10.* First, we show that there are at most 2 phases where the break from the loop (i.e., $\min_p d_{(p,1)} = 1$) occurs. We first show that a break never occurs if

$$\Delta \le \frac{17}{10} \frac{\epsilon_p}{\rho}.$$

This holds because

$$\hat{\Delta}(p) \le \Delta + \mathrm{Conf}(p) \tag{36}$$

$$\le \frac{17}{10} \frac{\epsilon_p}{\rho} + \mathrm{Conf}(p) \tag{37}$$

$$\le \frac{2\epsilon_p}{\rho} \quad \left( \text{by } \frac{20 - 17}{10} \frac{\epsilon_p}{\rho} \ge \frac{6}{10} \frac{\epsilon_p}{1} \ge \frac{\epsilon_p}{C_\rho} = \mathrm{Conf}(p) \right) \tag{38}$$

$$\le \frac{(2 + U_p)\epsilon_p}{\rho}, \tag{39}$$

We next show that

$$\Delta \geq \frac{17}{5} \frac{\epsilon_p}{\rho}$$

implies a break. This holds since

$$\hat{\Delta}(p) \geq \Delta - \mathrm{Conf}(p) \quad (\text{by } \mathcal{G}) \tag{40}$$

$$\geq \frac{17}{5} \frac{\epsilon_p}{\rho} - \mathrm{Conf}(p) \tag{41}$$

$$\geq \frac{3\epsilon_p}{\rho} \quad (\text{by } \frac{(17-15)\epsilon_p}{5\rho} \geq \frac{4\epsilon_p}{5} \geq \frac{\epsilon_p}{C_\rho} \geq \mathrm{Conf}(p)) \tag{42}$$

$$\geq \frac{(2+U_p)\epsilon_p}{\rho}. \tag{43}$$

The above results, combined with the fact that $\epsilon_p$ is halved at each phase, implying that the only decision points where the decision variable can take both of $\{0, 1\}$ are $p_s, p_s + 1$. Therefore, if the decision variable at phase $p_s$ matches, the decision variable at $p_s + 1$ and subsequent match. □

*Proof of Lemma 11.* For each phase $p \in \{p_s, p_s + 1\}$, we bound the probability of non-replication. At the end of phase $p$, it utilizes the randomness $U_p \sim \mathrm{Unif}(0, 1)$, and the random variable $\frac{(2+U_p)\epsilon_p}{\rho}$ is uniformly distributed on a region on size $\epsilon_p/\rho$. Meanwhile, event $\mathcal{G}$ implies

$$\left| \left( \hat{\Delta}^{(1)}(p) - \frac{(2+U_p)\epsilon_p}{\rho} \right) - \left( \hat{\Delta}^{(2)}(p) - \frac{(2+U_p)\epsilon_p}{\rho} \right) \right| \leq 2\mathrm{Conf}(p). \tag{44}$$

This implication suggests that within a region of at most width $2\mathrm{Conf}(p)$, the expressions $\left( \hat{\Delta}^{(1)}(p) - \frac{(2+U_p)\epsilon_p}{\rho} \right)$ and $\left( \hat{\Delta}^{(2)}(p) - \frac{(2+U_p)\epsilon_p}{\rho} \right)$ can have different signs. Therefore, the probability of non-replication is at most

$$\frac{2\mathrm{Conf}(p)\rho}{\epsilon_p} \leq 8\rho/9,$$

where the last inequality follows from the assumption $C_\rho \geq 9/4$. □

## C.2 Regret bound of Algorithm 1

This section derives the regret bound in Theorem 3.

Assume that $\mathcal{G}$ holds. Lemma 10 implies that the break occurs by the end of phase $p_s$ or $p_s + 1$. The regret up to this phase is at most

$$\sum_i \Delta_i \times N_{p_s+1}.$$

By using equation 7, we have

$$N_{p_s+1} = O\left( \log T \times 4^{p_s} \right) = O\left( \log T \times \frac{1}{\Delta^2 \rho^2} \right).$$

Moreover, assume that $\hat{\mu}_i(p) \geq \hat{\mu}_1(p)$ when a break occurs at $p \in \{p_s, p_s + 1\}$. Then,

$$\hat{\mu}_i(p) - \hat{\mu}_1(p) \geq \frac{2\epsilon_{p_s}}{\rho} \quad (\text{by the fact that break occurs}) \tag{45}$$

$$\geq 2\epsilon_{p_s} \tag{46}$$

Meanwhile, $\mathcal{G}$ implies for all phase $p$

$$\hat{\mu}_i(p) - \hat{\mu}_1(p) \leq 2\mathrm{Conf}(p). \tag{47}$$

Since $\mathrm{Conf}(p) < \epsilon_p$ for all $p$, this contradicts. By proof of contradiction. $\hat{\mu}_1(p) > \hat{\mu}_i(p)$. Therefore, the empirically best arm is always the true best arm, and we have zero regret during the exploitation period under $\mathcal{G}$. In summary,

$$\mathbb{E}[\mathrm{Regret}(T)] \leq \mathbb{E}[\mathbf{1}[\mathcal{G}] \cdot \mathrm{Regret}(T)] + O(1) \quad (\text{equation 35 implies } \Pr[\mathcal{G}^c] \text{ is } K/T = O(1/T))$$

$$\tag{48}$$

$$\leq \qquad \underbrace{O\left(\sum_i \Delta_i N_{P_c}\right)}_{\text{Regret during exploration, due to Lemma 10}} + \underbrace{0}_{\text{Regret during exploitation}} + \underbrace{O(1)}_{\text{Regret in the case of } \mathcal{G}^c} \qquad (49)$$

$$\leq O\left(\sum_i \Delta_i \frac{\log T}{\Delta^2 \rho^2}\right) + O(1) \qquad (50)$$

$$= O\left(\sum_i \Delta_i \frac{\log T}{\Delta^2 \rho^2}\right), \qquad (51)$$

which completes the proof.

## D  PROOFS ON ALGORITHM 2

### D.1  REPLICABILITY OF ALGORITHM 2

Similar to that of Theorem 3, we utilize Theorem 2. We use the event $\mathcal{G}$ defined in equation 33 during the proof.

**Lemma 12.** *Event $\mathcal{G}$ holds with probability at least $1 - \rho/6$. Moreover, under $\mathcal{G}$, arm 1 (= best arm) is never eliminated (i.e., $1 \in \mathcal{A}_p$ for all $p$).*

There are $K + 1$ binary decision variables at phase $p$. The first decision variable is

$$d_{(p,0)} := \mathbf{1}\left[\hat{\Delta}(p) > \frac{(2 + U_{p,0})\epsilon_p}{\rho_a}\right], \qquad (52)$$

which corresponds to Line 7. The other $K$ decision variables are

$$d_{(p,i)} := \mathbf{1}\left[\hat{\Delta}_i(p) > \frac{(2 + U_{p,i})\epsilon_p}{\rho_e}\right] \qquad (53)$$

for each $i = 1, 2, \ldots, K$, which corresponds to Line 10. If all decision variables are identical between two runs then the sequence of draws is identical between them.

**Lemma 13.** *Let*

$$p_{\mathrm{s},0} = \min_p \left\{\epsilon_p \leq \frac{10\Delta}{17\rho_a}\right\}.$$

*Under $\mathcal{G}$, we have $\rho^{(p,0)} = 0$ for any $p \neq p_{\mathrm{s},0}$.*

**Lemma 14.** *Let*

$$p_{\mathrm{s},i} = \min_p \left\{\epsilon_p \leq \frac{10\Delta_i}{17\rho_e}\right\},$$

*for each $i \in [K]$. Under $\mathcal{G}$, we have $\rho^{(p,1)}, \rho^{(p,2)} = 0$ for all $p$. Moreover, $\rho^{(p,i)} = 0$ for all $p \neq p_{\mathrm{s},i}$.*

**Lemma 15.** *Under $\mathcal{G}$, the probability of non-replication at each decision point is at most $\rho^{(p,0)} = 8\rho_a/9$ or $\rho^{(p,i)} = 8\rho_e/9$ for $i \in \{3, \ldots, K\}$.*

*Proof of non-replicability part of Theorem 4.* Theorem 2 and Lemmas 12–15 imply that the probability of misidentification is at most $2 \times \rho/18 + 8(\rho_a + (K-2)\rho_e)/9 = \rho$, which completes the proof. $\qquad \square$

*Proof of Lemma 12.* Lemma 9 implies that event $\mathcal{G}$ holds with probability at least $1 - \rho/6$.

In the following, we show that arm 1 is never eliminated under $\mathcal{G}$. Elimination of arm 1 at phase $p$ at Line 7 implies that there exists a suboptimal arm $i \neq 1$ such that

$$\hat{\mu}_i(p) - \mu_1(p) \geq \frac{(2 + U_{p,0})\epsilon_p}{\rho_a} \geq \frac{2\epsilon_p}{\rho_a} \geq 2\epsilon_p,$$

which never occurs under $\mathcal{G}$, since $\mathcal{G}$ implies

$$|\mu_1 - \hat{\mu}_1(p)|, |\mu_i - \hat{\mu}_i(p)| \leq \mathrm{Conf}(p) < \epsilon_p/2$$

and $\mu_1 > \mu_i$ by definition. The same discussion goes for elimination at Line 10. $\qquad \square$

*Proof of Lemma 13.* We first show arm 2 is never eliminated by Line 10 before phase $p_{s,0} + 1$. For $p \le p_{s,0} + 1$, we have

$$\hat{\Delta}_2(p) - \frac{(2 + U_{p,l})\epsilon_p}{\rho_e} \le \hat{\Delta}_2(p) - \frac{2\epsilon_p}{\rho_e} \tag{54}$$

$$\le \Delta + 2\mathrm{Conf}(p) - \frac{2\epsilon_p}{\rho_e} \quad \text{(by } \mathcal{G}) \tag{55}$$

$$= \Delta + \frac{2\epsilon_p}{C_\rho} - \frac{2\epsilon_p}{\rho_e} \tag{56}$$

$$\le \frac{17\epsilon_p}{5\rho_a} + \frac{2\epsilon_p}{C_\rho} - \frac{2\epsilon_p}{\rho_e} \quad \text{(by } p \le p_{s,0}) \tag{57}$$

$$\le \frac{17\epsilon_p}{5\rho_a} + \frac{2\epsilon_p}{3} - \frac{2\epsilon_p}{\rho_e} \tag{58}$$

$$\le 0, \quad \text{(by } 1/2 \ge \rho_a \ge 2\rho_e) \tag{59}$$

which implies $d_{(p,2)} = 0$.

Given arm 1 is never eliminated and arm 2 remains in $\mathcal{A}_{p_{s,0}+1}$ unless $d_{(p,0)} = 1$ occurs, the minimum gap $\Delta = \mu_1 - \mu_2$ is the same as the minimum gap among $\mathcal{A}_{p_{s,0}+1}$. The rest of this Lemma, which shows $d_{(p,0)} = 1$ for the first phase at $p = p_{s,0}$ or $p_{s,0} + 1$, is very similar to that of Lemma 10, and thus we omit it. The only non-zero decision variable is $\rho^{(p,0)}$ at $p = p_{s,0}$ by using the same discussion. $\qquad\square$

*Proof of Lemma 14.* We omit the proof because it is very similar to Lemma 10. $\qquad\square$

*Proof of Lemma 15.* We omit the proof because the proof for each $i = 0, 1, 2, \ldots, K$ is very similar to Lemma 11.

$\qquad\square$

### D.2 REGRET BOUND OF ALGORITHM 2

**(A) The $O(K)$ regret bound:** We first derive the same bound as Algorithm 1. This part is very identical to that of Algorithm 1 because, under $\mathcal{G}$, all but arm 1 is eliminated by phase $p_{s,0} + 1$.

**(B) The other two regret bounds:** We first derive the distribution-dependent bound. Lemma 15 states that, under $\mathcal{G}$, each arm $i$ is eliminated by phase $p_{s,i} + 1$. Therefore, each arm $i$ is drawn at most

$$N_{i,\max} := O\left(4^{p_{s,i}+1} \log T\right) = O\left(\frac{K^2}{\rho^2 \Delta_i^2} \log T\right) \tag{60}$$

times, and thus regret is bounded as

$$\mathbb{E}[\mathrm{Regret}(T)] \le \mathbb{E}[\mathbf{1}[\mathcal{G}] \cdot \mathrm{Regret}(T)] + O(1) \tag{61}$$

$$\le \underbrace{\sum_i \Delta_i N_{i,\max}}_{\text{Regret during exploration}} + \underbrace{0}_{\substack{\text{Regret during exploitation, which is 0 by Lemma 12}}} + \underbrace{O(1)}_{\substack{\text{Regret in the case of } \mathcal{G}^c}} \tag{62}$$

$$\le O\left(\sum_i \frac{K^2 \log T}{\Delta_i \rho^2}\right), \tag{63}$$

which is the second regret bound of Theorem 4.

We finally derive the distribution-independent regret bound. Letting $N_i(T)$ be the number of draws of arm $i$ in the $T$ rounds, we have

$$\mathrm{Regret}(T)\mathbf{1}[\mathcal{G}] \le \sum_i \Delta_i N_i(T) + O(1) \tag{64}$$

$$= \sum_i \Delta_i \sqrt{N_{i,\max}} \sqrt{N_i(T)} + O(1), \tag{65}$$

and

$$\sum_i \Delta_i \sqrt{N_{i,\max}} \sqrt{N_i(T)} \leq O(1) \times \sum_i \Delta_i \sqrt{\frac{K^2}{\rho^2 \Delta_i^2} \log T} \sqrt{N_i(T)} \quad \text{(by equation 60)} \tag{66}$$

$$\leq O(1) \times \sum_i \sqrt{\frac{K^2}{\rho^2} \log T} \sqrt{N_i(T)} \tag{67}$$

$$\leq O(1) \times \sqrt{\frac{K^2}{\rho^2} \log T} \sqrt{KT} \quad \text{(by Cauchy-Schwarz and } \sum_i N_i(T) = T\text{),} \tag{68}$$

and thus

$$\mathbb{E}[\mathrm{Regret}(T)] \leq \mathbb{E}[\mathrm{Regret}(T)\mathbf{1}[\mathcal{G}]] + O(1) \tag{69}$$

$$= O\left(\sqrt{\frac{K^2}{\rho^2} \log T} \sqrt{KT}\right) \quad \text{(by equation 106)} \tag{70}$$

$$= O\left(\frac{K}{\rho} \sqrt{KT \log T}\right), \tag{71}$$

which is the third regret bound of Theorem 4.

## E    PROOFS ON THE LOWER BOUND

In the following, we derive Theorem 5. The proof is inspired by Theorem 7.2 of Impagliazzo et al. (2022) but is significantly more challenging due to the adaptiveness of sampling. In particular, Lemma 16 utilizes the change-of-measure argument and works even if the number of draws $N_2(T)$ on arm 2 is a random variable.

*Proof of Theorem 5.* The goal of the proof here is to derive the inequality:

$$\mathbb{E}[\mathrm{Regret}(T)] = \Omega\left(\frac{1}{\rho^2 \Delta \log((\rho\Delta)^{-1})}\right). \tag{72}$$

We consider the set of models $\mathcal{P}$, where $\mu_1 = 1/2$ is fixed and $\mu_2 \in [1/2 - \Delta, 1/2 + \Delta]$. With a slight abuse of notation, we specify a model in $\mathcal{P}$ by $\mu_2 - 1/2$. We also denote $U$ to the internal randomness For example,

$$\mathbb{P}_{-\Delta,U}[\mathcal{X}]$$

be the probability that event $\mathcal{X}$ occurs under the corresponding model $(\mu_1, \mu_2) = (1/2, 1/2 - \Delta)$ and randomness $U$.

Markov's inequality on the regret and the assumption of uniformly goodness implies that, there exists a sublinear function $R(T)$ such that, at least 3/4 of the choice of the randomness $U$, we have $\mathbb{E}_{-\Delta,U}[\mathrm{Regret}(T)] \leq R(T)$. Similarly, at least 3/4 of the choice of the randomness $U$, we have $\mathbb{E}_{\Delta,U}[\mathrm{Regret}(T)] \leq R(T)$. By taking a union bound, at least half $(= 1 - (1/4 + 1/4))$ over the choice of $U$, we have

$$\max\{\mathbb{E}_{-\Delta,U}[\mathrm{Regret}(T)], \mathbb{E}_{\Delta,U}[\mathrm{Regret}(T)]\} \leq R(T). \tag{73}$$

Let $T_0$ be such that

$$\forall T \geq T_0 \ R(T) \leq \frac{T\Delta}{8}. \tag{74}$$

In the following, we consider an arbitrary $T \geq T_0$. In parts (A) and (B), we fix $U$ such that equation 73 holds. Part (C) marginalizes it over $U$.

**(A) Fix the randomness $U$ and consider behavior of algorithm for different models:**

Let $N_C$ be the smallest among the integer $n$ such that

$$\mathbb{P}_{-\Delta, U}\left[N_2(T) \geq n\right] \leq \frac{1}{4} \text{ and} \tag{75}$$

$$\mathbb{P}_{\Delta, U}\left[N_2(T) \geq n\right] \geq \frac{3}{4}. \tag{76}$$

At least one such an integer exists; $n = T/2$ satisfies this condition because otherwise

$$\max\{\mathbb{E}_{-\Delta, U}[\text{Regret}(T)], \mathbb{E}_{\Delta, U}[\text{Regret}(T)]\} \tag{77}$$

$$\geq \max\left\{\mathbb{P}_{-\Delta, U}\left[N_2(T) \geq \frac{T}{2}\right]\frac{T\Delta}{2}, \left(1 - \mathbb{P}_{\Delta, U}\left[N_2(T) \geq \frac{T}{2}\right]\right)\frac{T\Delta}{2}\right\} \tag{78}$$

$$> \frac{T\Delta}{8} \tag{79}$$

$$\text{(by } \mathbb{P}_{-\Delta, U}\left[N_2(T) \geq \frac{T}{2}\right] > \frac{1}{4} \text{ or } \mathbb{P}_{\Delta, U}\left[N_2(T) \geq \frac{T}{2}\right] < \frac{3}{4}), \tag{80}$$

which violates equation 74.

By continuity, there exists $\nu = \nu(U) \in (-\Delta, +\Delta)$ such that $\mathbb{P}_{\nu, U}[N_2(T) \geq N_C] = 1/2$. By Lemma 16, there exists $C_I = \Theta(1)$ such that, for any $\nu' \in [\nu - C_I/\sqrt{\log(N_C)N_C}, \nu + C_I/\sqrt{\log(N_C)N_C}] =: \mathcal{I}(U)$ we have $\mathbb{P}_{\nu', U}[N_2(T) \geq N_C] \in (1/3, 2/3)$.

**(B) Marginalize it over models:** Assume that we first draw a model uniformly random from $[-\Delta, \Delta]$, and then run the algorithm. Conditioned on the shared randomness $U$, with probability at least $C_I/(\sqrt{\log(N_C)N_C}\Delta)$, we draw model in $\mathcal{I}(U)$. For a model in $\mathcal{I}(U)$, there is $2 \times (1/3) \times (2/3) = 4/9$ probability of non-replicability.

**(C) Marginalize it over shared random variable $U$:** equation 73 implies that the discussions in (A) and (B) hold at least half of the random variable $U$. Marginalize it over the distribution on $U$, we have the probability of non-replicability at least

$$\frac{1}{2} \times \frac{4}{9}\frac{C_I}{\sqrt{\log(N_C)N_C}\Delta}. \tag{81}$$

Since the algorithm is $\rho$-replicable, we have

$$\frac{2}{9}\frac{C_I}{\sqrt{\log(N_C)N_C}\Delta} \leq \rho,$$

which implies

$$N_C = \Omega\left(\frac{1}{(\rho\Delta)^2 \log((\rho\Delta)^{-1})}\right). \tag{82}$$

The regret is lower-bounded as

$$\max\{\mathbb{E}_{-\Delta}[\text{Regret}(T)], \mathbb{E}_{\Delta}[\text{Regret}(T)]\} \tag{83}$$
$$\geq \max\left\{\mathbb{P}_{-\Delta}[N_2(T) \geq N_C - 1]\Delta(N_C - 1), (1 - \mathbb{P}_{\Delta}[N_2(T) \geq N_C - 1])\Delta(T - N_C + 1)\right\} \tag{84}$$

$$\geq \max\left\{\mathbb{P}_{-\Delta}[N_2(T) \geq N_C - 1]\Delta(N_C - 1), (1 - \mathbb{P}_{\Delta}[N_2(T) \geq N_C - 1])\Delta\left(\frac{T}{2} + 1\right)\right\} \tag{85}$$

$$\text{(by } N_C \leq T/2) \tag{86}$$

$$\geq \frac{1}{4} \times \Delta(N_C - 1) \tag{87}$$

$$\text{(by } N_C - 1 \text{ violates equation 75 or equation 76, and } N_C \leq T/2) \tag{88}$$

$$= \Omega\left(\frac{1}{\rho^2\Delta \log((\rho\Delta)^{-1})}\right). \quad \text{(by equation 82)} \tag{89}$$

$\square$

### E.1 Lemmas for Regret Lower Bound

The following lemma is used to bound the gradient of the probability of occurrences. This lemma corresponds to the derivative of the acceptance function[4] of Lemma 7.2 in Impagliazzo et al. (2022), but more technical due to the fact that $N_2(T)$ is a random variable.

**Lemma 16.** (Likelihood ratio) *Let*

$$\mathcal{E} = \{N_2(T) \le N_C\} \tag{90}$$

*and $\nu$ be such that $\mathbb{P}_{\nu,U}[\mathcal{E}] = 1/2$. There exists a value $C_I = \Theta(1)$ that does not depend on the shared random variable $U$ such that, for any model $\nu' \in [\nu - C_I/\sqrt{\log(N_C)N_C}, \nu + C_I/\sqrt{\log(N_C)N_C}]$, we have*

$$\mathbb{P}_{\nu',U}[\mathcal{E}] \in \left(\frac{1}{3}, \frac{2}{3}\right). \tag{91}$$

*Proof of Lemma 16.* In the following, we bound $\mathbb{P}_{\nu',U}[\mathcal{E}]$ by using the change-of-measure argument.

Let the log-likelihood ratio between from models $\nu, \nu'$ be[5]

$$L_t = \sum_{n=1}^{N_2(t)} \log\left(\frac{X_{2,n}\left(\frac{1}{2}+\nu\right) + (1-X_{2,n})\left(\frac{1}{2}-\nu\right)}{X_{2,n}\left(\frac{1}{2}+\nu'\right) + (1-X_{2,n})\left(\frac{1}{2}-\nu'\right)}\right), \tag{92}$$

where $X_{2,n}$ is the $n$-th reward from arm 2 and $N_2(t)$ be the number of draws on arm 2 during the first $t$ rounds.

We have

$$\mathbb{P}_{\nu',U}[\mathcal{E}] = \mathbb{E}_{\nu,U}\left[\mathbf{1}[\mathcal{E}]e^{-L_T}\right]. \quad \text{(change-of-measure)} \tag{93}$$

Note that, under $\nu$ the random variable

$$\log\left(\frac{X_{2,n}\left(\frac{1}{2}+\nu\right) + (1-X_{2,n})\left(\frac{1}{2}-\nu\right)}{X_{2,n}\left(\frac{1}{2}+\nu'\right) + (1-X_{2,n})\left(\frac{1}{2}-\nu'\right)}\right)$$

is mean $d_{\mathrm{KL}}(\nu, \nu')$ and bounded by

$$R = \left|\log\left(\frac{2+\nu}{2+\nu'}\right), \log\left(\frac{2-\nu}{2-\nu'}\right)\right| = O(|\nu - \nu'|) = O\left(\frac{C_I}{\sqrt{\log(N_C)N_C}}\right).$$

Under $\mathcal{E}$, $N_2(T) \le N_C$ and $L_T$ is bounded as the max of random variables

$$L_T \le \max_{N \le N_C}\left(\sum_{n \le N} Z_n\right),$$

where

$$Z_n := \log\left(\frac{X_{2,n}\left(\frac{1}{2}+\nu\right) + (1-X_{2,n})\left(\frac{1}{2}-\nu\right)}{X_{2,n}\left(\frac{1}{2}+\nu'\right) + (1-X_{2,n})\left(\frac{1}{2}-\nu'\right)}\right)$$

is a random variable with its mean $d_{\mathrm{KL}}(\nu, \nu')$ and radius $R$. Hoeffding inequality and union bound over $N = 1, 2, \ldots, N_C$ implies that, with probability at least $1 - 1/12$ we have

$$|L_T| \le N_C d_{\mathrm{KL}}(1/2 + \nu, 1/2 + \nu') + R\sqrt{\log(2 \times 12 \times N_C)N_C/2} \tag{94}$$

$$= O\left(N_C \times \frac{C_I^2}{\log(N_C)N_C}\right) + O\left(\frac{C_I}{\sqrt{\log(N_C)N_C}} \times \sqrt{N_C \log(N_C)}\right) = O(C_I) \tag{95}$$

and by setting an appropriate width $C_I = \Theta(1)$ guarantees that $e^{-L_T} \in [1 - 1/6, 1 + 1/6]$ with probability at least $1 - 1/12$, which, together with the change-of-measure implies equation 91. $\square$

---

[4]Namely, $\mathrm{ACC}(p)$ therein.
[5]On these models, $\mu_1 = 1/2$, $\mu_2 = 1/2 + \nu$ or $1/2 + \nu'$.

# F PROOFS ON ALGORITHM 3

## F.1 REPLICABILITY OF ALGORITHM 3

We omit the derivation of the non-replicability bound of Algorithm 3 because it is very similar to that of Algorithm 2. The only difference is that the amount of exploration is based on a G-optimal design, but its confidence bound of equation 19 suffices to derive the good event that is identical to equation 33.

## F.2 REGRET BOUND OF ALGORITHM 3

This section shows the regret bounds of Theorem 8. The main difference from Algorithm 2 is that the number of samples for each arm is $N_i^{\mathrm{lin}}(p)$ that satisfies $\sum_i N_i^{\mathrm{lin}}(p) \leq N^{\mathrm{lin}}(p) + K$.

**(A) The $O(d)$ regret bound:** We derive the first regret bound in Theorem 8. We first derive the distribution-dependent bound. Similar discussion as Lemma 15 states that, under $\mathcal{G}$, Line 7 in Algorithm 3 eliminates all but best arm by phase $p_{\mathrm{s},0} + 1$, where $p_{\mathrm{s},0}$ is identical to that of Algorithm 3. The regret is bounded as

$$\mathbb{E}[\mathrm{Regret}(T)] \leq \mathbb{E}[\mathbf{1}[\mathcal{G}]\mathrm{Regret}(T)] + O(1) \tag{96}$$

$$\leq \underbrace{O\left(\sum_i N_i^{\mathrm{lin}}(p_{\mathrm{s},i}+1)\right)}_{\text{Regret during exploration}} + \underbrace{0}_{\text{Regret during exploitation}} + \underbrace{O(1)}_{\text{Regret in the case of }\mathcal{G}^c} \tag{97}$$

$$= O\left(\frac{d\log T}{\Delta^2 \rho^2}\right) + o(\log T), \tag{98}$$

$$\text{(by } \sum_i N_i^{\mathrm{lin}}(p_{\mathrm{s},i}+1) = N^{\mathrm{lin}}(p_{\mathrm{s},i}+1) \leq \frac{16\sigma^2 d\log(|\mathcal{A}_p|PT)}{(\mathrm{Conf}(p_{\mathrm{s},0}))^2} = O\left(\frac{d\log T}{\Delta^2 \rho^2}\right)) \tag{99}$$

which is the first regret bound of Theorem 8.

**(B) The distribution-independent regret bound:**

Similar discussion as Algorithm 2 states that, under $\mathcal{G}$, arm $i$ is eliminated by $p_{\mathrm{s},i} + 1$. The regret is bounded as

$$\mathrm{Regret}(T)\mathbf{1}[\mathcal{G}] \leq \sum_i \Delta_i N_i(T) + O(1) \tag{100}$$

$$= \sum_i \Delta_i \sqrt{\sum_{p \leq p_{\mathrm{s},i}+1} N_i^{\mathrm{lin}}(p)} \sqrt{N_i(T)} + O(1), \tag{101}$$

and

$$\sum_i \Delta_i \sqrt{\sum_{p \leq p_{\mathrm{s},i}+1} N_i^{\mathrm{lin}}(p)} \sqrt{N_i(T)} \tag{102}$$

$$\leq O(1) \times \sum_i \Delta_i \sqrt{\frac{dK^2}{\rho^2 \Delta_i^2} \log T} \sqrt{N_i(T)} \quad \text{(by equation 17)} \tag{103}$$

$$\leq O(1) \times \sum_i \sqrt{\frac{dK^2}{\rho^2} \log T} \sqrt{N_i(T)} \tag{104}$$

$$\leq O(1) \times \sqrt{\frac{dK^2}{\rho^2} \log T} \sqrt{KT} \quad \text{(by Cauchy-Schwarz and } \sum_i N_i(T) = T), \tag{105}$$

$$\tag{106}$$

and thus

$$\mathbb{E}[\mathrm{Regret}(T)] \leq \mathbb{E}[\mathrm{Regret}(T)\mathbf{1}[\mathcal{G}]] + O(1) \tag{107}$$

$$= O\left(\sqrt{\frac{dK^2}{\rho^2}\log T}\sqrt{KT}\right) \quad \text{(by equation 106)} \tag{108}$$

$$= O\left(\frac{K}{\rho}\sqrt{dKT\log T}\right), \tag{109}$$

which is the second regret bound of Theorem 8.

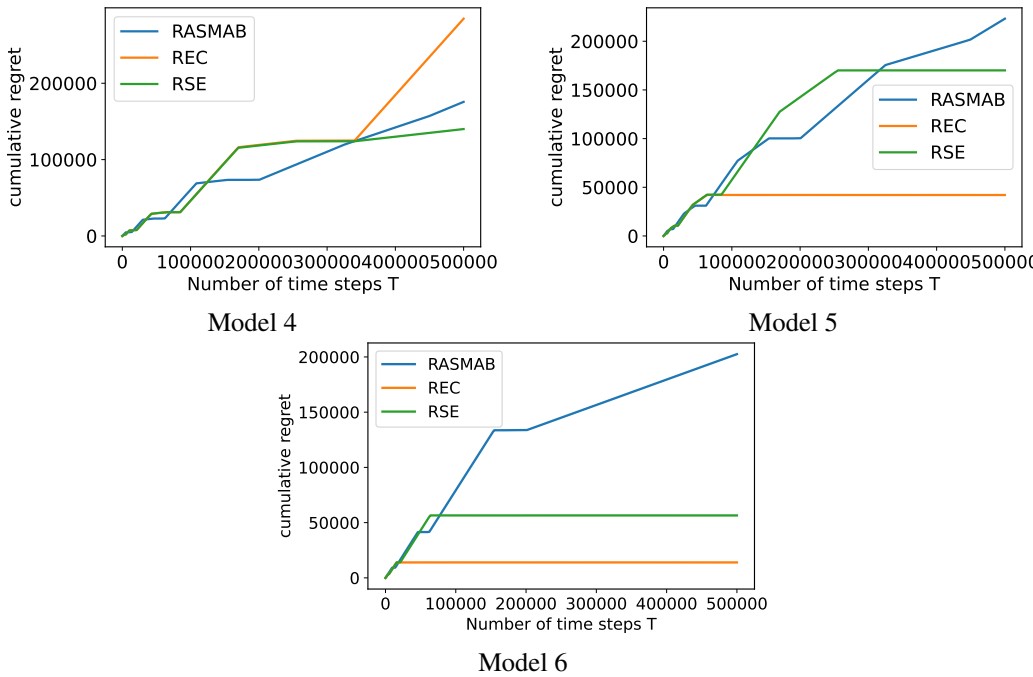

Figure 2: Regret of algorithms with theoretical parameters.

## G  ADDITIONAL SIMULATION

In this section, we report simulation results in which the algorithmic hyperparameters are selected based on their replicability bounds. Specifically, we set $C_\rho = 9/4$ for REC and RSE, and we adhere to the original version of Algorithm 2 as outlined in Esfandiari et al. (2023a). It should be noted that RASMAB is not explicitly designed to meet a non-replication level of $\rho$, owing to its disregard of the multiplication factor on the number of effective decision variables. This fact suggests that this comparison offers a substantial advantage to RASMAB . Notably, Appendix B therein specifies that there are 6 possible bad events. This implies that, for large $T$, their algorithm would need to choose an amount of exploration at least $(6-1)^2$ times greater.

We have set the models (i.e., mean parameters) as follows: $\boldsymbol{\mu} = (0.1, 0.9, 1.0)$ for Model 4, $\boldsymbol{\mu} = (0.1, 0.5, 1.0)$ for Model 5, and $\boldsymbol{\mu} = (0.0, 0.0, 1.0)$ for Model 6. Additionally, we set $\rho = 0.5$. Model 4 is more advantageous for RASMAB and RSE, while Model 6 favors REC.

The results for the Theoretical configuration are presented in Figure 2. All algorithms were confirmed to be $\rho$-replicable. However, due to the conservative selection of theoretical hyperparameters, the required exploration amount exceeded that of previous simulations. As a result, all algorithms incurred a larger regret compared to the simulations in the main paper.

Among the three algorithms, REC showed superior performance in Models 5 and 6, while RASMAB and RSE had some advantages in Model 4. These results align with our theoretical findings, which suggest that REC and RSE have a significant advantage over RASMAB when the value of $\Delta_K/\Delta$ is moderate.