# OpenReview forum: "Improved Algorithms for Replicable Bandits"
_ICLR.cc/2024/Conference — Submitted to ICLR 2024_

### Official Review · Reviewer_apnD · 2023-10-25

**Soundness:** 2 fair
**Presentation:** 3 good
**Contribution:** 2 fair
**Rating:** 6
**Confidence:** 4

**Summary:**

This paper designs replicable algorithms for multi-armed bandits and linear bandits, i.e., with a fixed inner random variable, the vector of the pulled arms will remain the same with high probability, even if the outcomes of arms are also random. The authors adapt the RASMAB algorithm framework in Esfandiari et al. (2023), and modify the way of eliminating arms to get new algorithms REC and RSE. In REC, the algorithm only tries to eliminate all the arms in one single phase. This leads to an $O(\sum_i {\Delta_i \log T \over \Delta_{\min}^2 \rho^2})$ regret upper bound, which is better than existing algorithms when $\Delta_{\min} \ge \Delta_i/K$ for all arm $i$. Then the authors mix REC and RASMAB to get RSE, which achieves a regret upper bound that equals the minimum of the regret upper bounds of REC and RASUCB. As for the linear bandits case, the authors also provide an RLSE algorithm. Finally, they use some simulation results to demonstrate the effectiveness of their algorithms.

**Strengths:**

The paper is clearly written, and there are some new regret upper/lower bounds.

**Weaknesses:**

My main concern is that the writing of proofs is not good enough. In fact, I only check one proof in the appendix (Theorem 5), but I find it hard to understand some of the steps.

Specifically, Eq. (75) and Eq. (76) only say that $P_{-\Delta, U}[N_2(T) \ge N_C] \ge {1\over 4}$ and $P_{\Delta, U}[N_2(T) \ge N_C] \ge {3\over 4}$. This cannot ensure that there exists some $r \in [-\Delta, \Delta]$ such that $P_{r, U}[N_2(T) \ge N_C] = {1\over 2}$. For example, perhaps $P_{r, U}[N_2(T) \ge N_C] = 1$ for all $r \in [-\Delta, \Delta]$.

There are also many typos in the proof (which makes it more complicated to understand the steps), e.g., in the line after Eq. (75), I think it should be $Regret_{-\Delta,U}(T) \ge N_2(T)\Delta$ but not $Regret_{-\Delta,U}(T) \ge N_2(T)\Delta/4$, and it should be $N_C \le {T\over 2}$ but not $N_2(T) \le {T\over 2}$.

I really suggest the authors revise their proofs to make sure that they are correct and easy to understand.

Also, I think some of the proof sketches should be given in the main text.

Besides, both the analysis and experiments show that REC (or RSE) does not dominate RASMAB, and I am also wondering the experiments of performance comparison between RLSE and existing benchmarks.

=============================

I would like to raise my score if my concern on the proof is addressed.


=============================

The proof after revision looks clear now. I change my score to 6.

**Questions:**

See the above "Weaknesses"

---

> ### Author Response · Authors · 2023-11-17
> **On the error of the lower bound proof**
>
> Thank you for your careful reading and for correctly pointing out our mistake. We fixed the proof in the revision by defining $N_C$ to be the minimum of a set of integers (Appendix E, changes are colored red). Let us know if this is unclear.
>
> > Also, I think some of the proof sketches should be given in the main text.
>
> Thank you for your suggestion. We could not put them due to page limitation.

---

### Official Review · Reviewer_Hx4P · 2023-10-31

**Soundness:** 3 good
**Presentation:** 3 good
**Contribution:** 2 fair
**Rating:** 3
**Confidence:** 3

**Summary:**

This work analyzed the replicable bandits. The replicable bandit algorithms were first analyzed by Esfandiari et al. (2023). This work proposed the REC and RSE algorithm to solve the regret minimization problem in the standard replicable bandits and also studied the linear setting. It provided theoretical analysis on for the proposed algorithm and numerical comparison among algorithms.


==========

Thanks for response from author(s). I prefer to keep my score so far.

**Strengths:**

1. The paper is generally well organized and easy to follow.
1. The problem is clearly formulated.
2. Numerical experiments are conducted to evaluate the performance of algorithms.
1. This work provides extensive study on the replicable bandits.

**Weaknesses:**

Title:
1. The title of this work starts with 'Improved algorithms'. I doubt whether it is an appropriate title. I feel that this work is more likely to be an extensive study on replicable bandits.
    1. According to Table 1, the REC and RSE algorithms are only better than the existing RASMAB in the instances where $K^2/\Delta^i$ is larger than $\Delta^2_i/\Delta$. In other cases, the order of distribution-dependent upper bounds of these three algorithm seem to be the same.
    1. Numerical results imply that RASMAB algorithm outperforms the REC and RSE algorithms in Model 1. I appreciate some explanation regrading this point.

Application:
1. I understand it is a follow-up work on the topic of replicable bandits. However, if the aim is to handle situations where we have simply vague data, I think we can also apply differentially private stochastic bandits (https://proceedings.mlr.press/v180/hu22a/hu22a.pd). May you clarify the difference between application of these two settings? What are their individual strengths/weaknesses?

Contributions:
1. The distribution-independent bound is only provided for the RSE algorithm. However, it is usually not so difficult to derive a distribution-independent bound for the other algorithms when the distribution-dependent ones already exist. I suggest the author(s) to provide distribution-independent bounds for other algorithms and a comparison or explain the analytical challenge.
1. As the lower bound for the standard setting is provided, may the author(s) discuss the gap between upper and lower bounds?
1. I appreciate the efforts to study the linear setting, may the author(s) also provide a lower bound (or explain the difficulty)?
1. As the linear algorithm is proposed for an instance with a large number of arms, may the author(s) compare(s) the performance of linear and nonlinear algorithms in a numerical experiment with a large $K$?
1. Section 1.2 claimed that 'we introduce the Replicable Successive Elimination (RSE) algorithm whose regret bound is the minimum
of those of REC and the existing algorithms.'
    1. From Table 1, I cannot tell RSE is obviously better than REC.
    1. According to experiments, REC seems to be always slightly better than RSE.
    1. If RSE is better than REC, why are we interested in the REC algorithm?

Minor suggestion:
1. I appreciate the comparison among bounds in Table 1 but have a minor suggestion: it would be better to include the name of algorithms( and the theorems index), and rearrange the table.

**Questions:**

Please refer to the **Weaknesses** section for questions.

---

> ### Author Response · Authors · 2023-11-17
> **Thank you for your review**
>
> Thank you for your time and reading. We feel the review score is a bit harsh given the reviewer agrees on the novelty of the paper. It would be great if you raised the score to match the actual sentiment.
>
> > The title of this work starts with 'Improved algorithms'. I doubt whether it is an appropriate title. I feel that this work is more likely to be an extensive study on replicable bandits
>
> We agree with this. In particular, Section 3 formalizes the way to guarantee the reproducibility that is implicitly (and inprecisely in terms of the multiplication count) used in (Esfandiari et al.'23). We revised the paper so that this contribution is clear. We have not changed the title, but we welcome your suggestion.
>
> > Bingshan Hu, Nidhi Hegde. "Near-optimal Thompson sampling-based algorithms for differentially private stochastic bandits." UAI 2022.
>
> Thank you for pointing out relevant work. DP considers the change of decision against the change of a single data point, whereas in our case, we have more than one change of data points between two datasets that are generated from the identical data-generating process. We added this to the revision of the related work section (Section 9, changes are colored red).
>
> > The distribution-independent bound is only provided for the RSE algorithm. However, it is usually not so difficult to derive a distribution-independent bound for the other algorithms when the distribution-dependent ones already exist. I suggest the author(s) to provide distribution-independent bounds for other algorithms and a comparison or explain the analytical challenge.
>
> Thank you for your suggestion. Following your recommendation, we removed the claim to be "the first distribution-independent bound".
>
> > As the lower bound for the standard setting is provided, may the author(s) discuss the gap between upper and lower bounds?
>
> Thank you for clarifcation. The gap is tight up to a logarithmic factor for $K=2$. We added this in Section 6 (colored red).
>
> > I appreciate the efforts to study the linear setting, may the author(s) also provide a lower bound (or explain the difficulty)?
>
> Thank you for pointing it out. The difficulty here is not from the linear model but from extending the lower bound to $K>2$ arms. Note that even $K=2$, ours is the first lower bound in this problem.
>
> > From Table 1, I cannot tell RSE is obviously better than REC.
>
> You are completely right, for most practical cases REC is better than RSE. If $\Delta_2$ is very small compared with $\Delta_3,...,\Delta_K$, then RSE would outperform, which is model 1. In model 1, RSE eliminated arms $3--5$ after $T=1000$ and thus outperforms REC, but the difference looks subtle due to the exploration before $T<1000$.
>
> > I appreciate the comparison among bounds in Table 1 but have a minor suggestion: it would be better to include the name of algorithms( and the theorems index), and rearrange the table.
>
> Revised accordingly; thank you for your suggestion.
>
> Thank you for many other suggestions. We highlighted the changes in the red lines.

---

> > ### Comment · Reviewer_Hx4P · 2023-11-22
> > **Thanks for your response**
> >
> > I appreciate the response from the author(s). Still, some of my concerns are not resolved:
> > 1. I still feel that is not that appropriate to have the title of this work start with 'Improved algorithms'. Again, I suggest the author(s) to revise the title.
> > 2. The similarity and difference between the replicable bandits and DP bandits should be clarified in the section of 'related works'.
> > 3. As I mentioned before, I suggest the author(s) to provide distribution-independent bounds for other algorithms and a comparison or explain the analytical challenge.
> >
> > Besides, I wonder if the concern of Reviewer tkZN is well resolved.
> >
> > Considering the current situation, I may keep my score now.

---

> > > ### Author Response · Authors · 2023-11-22
> > > **Re: Thanks for your response**
> > >
> > > Thank you for your comments. Forgive us to address your concerns quickly:
> > >
> > > * We agree; we will change the title to include the extensiveness in the sense that it provide a principled framework for guaranteeing replicability level.
> > > * We added the statement that we commented in the current revision (revision 2). Thank you for pointing out our mistake in revision 1.
> > > * Please remove this point from the list of our contributions.

---

### Official Review · Reviewer_a2Sd · 2023-11-01

**Soundness:** 3 good
**Presentation:** 2 fair
**Contribution:** 2 fair
**Rating:** 6
**Confidence:** 5

**Summary:**

This paper considers the problem of replicability in the context of stochastic bandits. Following the definition of Impagliazzo et al. '22, and its adaptation to the bandit setting of Esfandiari et al. '23, a bandit algorithm is called replicable if, with probability at least 1-$\rho$, it pulls the exact same sequence of arms when executed twice in the same bandit environment when its internal randomness is shared but the rewards of the arms are drawn independently across the two executions.

In the multi-armed bandit setting, the authors provide algorithms with an instance-dependent bound of $O(K)$, where $K$ is the number of different arms, and instance-independent bound of $O(K^{1.5})$. Moreover, they provide lower bounds which show that the dependence on the replicability parameter $\rho$ is tight.

Then, the authors consider the linear bandit setting and they provide algorithms whose instance-dependent regret scales as $O(d)$ and the instance-independent regret scales as $O(K^{1.5}d^{0.5})$.

**Strengths:**

-The paper presents an algorithm that gets $O(K)$ instance-dependent bound in the multi-armed bandit setting.

-The authors provide a best of both worlds type of algorithm that achieves the best bound between their algorithm and the one in Esfandiari et al. '23.

-An instance-independent regret analysis of the algorithm is provided.

-A lower bound on the regret is given which establishes the optimality on the dependence of the replicability parameter $\rho$. In my opinion, this is the most technically interesting part of the paper.

-Similar upper bounds are provided in the linear bandit setting. The instance-independent analysis provided in this work uses a different regret decomposition from Esfandiari et al. '23, which allows the authors to get rid of a factor $K/\rho$ in the regret bound.

**Weaknesses:**

After reading the abstract and the introduction of the paper, I was excited to learn more about the results. However, I think there are several issues that make the results less exciting than they appear to be.

Let me start by explaining the main technical challenges in this line of work on replicable algorithms. The bandit algorithms of Esfandiari et al. '23 and of this paper use multiple calls to the replicable mean estimation algorithm of Impagliazzo et al. '22. Let $N$ be the number of times this algorithm is called. Because of a union bound over the probability of non-replication, the sample complexity overhead scales as $O(1/(\rho/N)^2)$. For this reason, both Esfandiari et al. '23, and this paper using a batching approach and call the replicable mean estimation subroutine (or a randomized thresholding scheme that closely resembles it) at the end of each batch. The reason why Esfandiari et al. '23 incur an extra overhead of $O(K^2)$ in the regret compared to the non-replicable setting is, essentially, a union bound over the number of arms. Let me now move on to pointing specific weaknesses of the paper.

-Abstract: "Existing algorithms require a regret scale of $O(K^3)$, which increases much faster than the number of actions (or “arms”), denoted as K. We introduce an algorithm with a distribution-dependent regret of $O(K)$" -> both bounds are distribution-dependent, but are incomparable since the bound of Esfandiari et al. has better dependence on the distribution-dependent parameters than the one in the current paper. I think the way it is stated in the abstract is a bit confusing.

-Abstract: "Additionally, we propose an algorithm for the linear bandit with regret of $O(d)$, which is linear in the dimension of associated features, denoted as d, and it is independent of K." -> this bound is distribution-dependent, I think it should be mentioned.

-Abstract: "Our algorithms exhibit substantial simplicity compared to existing ones" -> I don't agree with the statement, I will elaborate shortly.

-Page 2: "Upon closer examination of the problem, we discovered that $K^2$ factor can be eliminated" -> as I mentioned before, the bounds are incomparable. I think at first read this sentence gives the impression that the regret bound is improved.

-Page 2-3: "Furthermore, due to the simplicity of the algorithm, we establish the first distribution-independent regret bound for the replicable K-armed bandit problem." -> this claim is imprecise. Esfandiari et al.'23 have already established distribution-independent bounds for linear bandits (without any restriction on the relationship between d, K), which is a more general setting than multi-armed bandits. As I said before, I don't agree with the statement that the algorithms are "simpler" either.

-Section 3: I believe that the non-replication framework is inspired by the long line of work on batched bandits and bandits with low adaptation, so I think some citations are needed. Moreover, the same batching framework was essentially used in Esfandiari et al. '23 (even though it wasn't stated as such, its essence remains the same). I don't the benefit of stating it as an abstract framework. I think the authors could move some of this discussion to the appendix and elaborate more on their proofs technique in the main body.

-Section 3: The set $|A_{p+1}|$ which I assume denotes the set of active arms is not defined (please correct me if I have missed it).

-Section 4: The explore-then-commit algorithm was mentioned in the warm-up section of Esfandiari et al. for the case where the suboptimality gap $\Delta$ is given to the designer and a sketch of the regret analysis was also provided. Essentially, Algorithm 2 of the current submission gets rid of the assumption that $\Delta$ is known by using the doubling-trick approach. I believe some discussion about it would be useful,

-Section 5: Essentially, this algorithm is a best of both worlds combination of Algorithm 2 and the replicable arm elimination algorithm of Esfandiari et al. '23. It is easy to see that if the algorithm detects a large gap between the two best arms, it just eliminates all of them except for the best, otherwise it performs the replicable arm elimination process of Esfandiari et al. '23. I think this comparison should be mentioned in the text. Given this discussion, I don't see where the simplicity of this algorithm compared to Esfandiari et al. '23 lies.

-Section 5: "Here, eliminating all but one arm is equivalent
to switching to the exploration period." -> exploitation.

-Section 5: "Moreover,
this algorithm is the first replicable algorithm that has a distribution-independent regret bound in the
K-armed bandit problem." -> As I mentioned before, this is not correct. Since the distribution-independent bounds for linear bandits of Esfandiari et al. hold even when $K = d$ they already imply distribution-independent regret bound for the K-armed bandit problem.

-Section 7: "Next, we consider the linear bandit problem, a special version of the K-armed bandit problem
where associated information is available." -> if $d = K$ then the linear bandit problem can express the multi-armed bandit problem, so maybe it should be emphasized that $d << K.$

-Section 7: "The main innovation here is to use the G-optimal design that
explores all dimensions in an efficient way." -> I think it should be mentioned that this innovation has already been done in the literature since it is a very well-known technique and is not an innovative element of the current submission.

-Lemma 7: The definition of $\hat{\theta}_p$ is missing. I think it's worth discussing that even though $\hat{\theta}_p$ is different across two executions, because of the different observed rewards, the algorithm is still replicable.

-Related work: Throughout the draft, the citation to Esfandiari et al. '23 is not the indented one. In fact, the citation to the correct paper is absent from the references.

-Conclusion: "This represents a significant advancement over existing algorithms." -> as I explained before, the bounds are not comparable.

-In general, the authors could try to add some proof sketches in the main body. If need be, they could try to shorten the discussion about the replicability framework, which I don't think adds much to the paper.

-Not a weakness, but relevant the the related works section: Some other references that could be useful:
"List and Certificate Complexities in Replicable Learning", Peter Dixon, A. Pavan, Jason Vander Woude, N. V. Vinodchandran
"Replicability and stability in learning", Zachary Chase, Shay Moran, Amir Yehudayoff
Both of these works study a different notion of replicability that has to do with the number of different outputs of a learning algorithm.

"Replicable Reinforcement Learning", Eric Eaton, Marcel Hussing, Michael Kearns, Jessica Sorrell
"Replicability in Reinforcement Learning", Amin Karbasi, Grigoris Velegkas, Lin F. Yang, Felix Zhou
Both of these papers study replicable algorithms for RL.

"Statistical Indistinguishability of Learning Algorithms", Alkis Kalavasis, Amin Karbasi, Shay Moran, Grigoris Velegkas
This paper provides a relaxation of the replicability definition and extends some of the equivalences shown in Bun et al. '23 to uncountable domains.

**Questions:**

-See weaknesses section.

-Instance-independent regret bound for linear vs. multi-armed bandits: for linear bandits the regret bound is $O(K/\rho \sqrt{dKT\log T})$ whereas for multi-armed bandits it is $O(K/\rho \sqrt{KT\log T})$. Isn't the former always worse even though there is more structure?

-Appendix F.2, proof of the (A) case. Isn't there an additive term O(K) missing in the regret bound?

---

> ### Author Response · Authors · 2023-11-17
> **Thank you for many comments**
>
> Thank you for your careful reading. We updated the paper accordingly (colored red in the revision). Below are our updates.
>
> > Incompativility of $O(K^3)$ and $O(K)$ bounds
>
> You are correct. We denote this several times, but revised the paper to clarify that "when the suboptimality gaps for each arm are within a constant factor".
>
> > We establish the first distribution-independent regret bound for the replicable $K$-armed bandit problem
>
> We agree that and drop this statement.
>
> > Section 3: I believe that the non-replication framework is inspired by the long line of work on batched bandits and bandits with low adaptation.
>
> Thank you for the reference. We added batched bandit papers in the related work (Section 9).
>
> > Section 4: The explore-then-commit algorithm was mentioned in the warm-up section of Esfandiari et al. for the case where the suboptimality gap is given to the designer and a sketch of the regret analysis was also provided. Essentially, Algorithm 2 of the current submission gets rid of the assumption that is known by using the doubling-trick approach.
>
> Thank you for the reference. We added this at the end of Section 4. Their algorithm is $O(T \sum_i \Delta_i)$, and our EtC clearly provides a better bound.
>
> > Section 5: Essentially, this algorithm is a best of both worlds combination of Algorithm 2 and the replicable arm elimination algorithm of Esfandiari et al. '23. It is easy to see that if the algorithm detects a large gap between the two best arms, it just eliminates all of them except for the best, otherwise it performs the replicable arm elimination process of Esfandiari et al. '23. I think this comparison should be mentioned in the text. Given this discussion, I don't see where the simplicity of this algorithm compared to Esfandiari et al. '23 lies.
>
> We agree with the word "simplicity" is somewhat imprecise. We think Section 3 in our paper clarified what is somewhat implicitly used in [Esfandiari et al. '23], and we revised the paper so that our contributions are more clearly mentioned.
>
> > Not a weakness, but relevant the the related works section: Some other references that could be useful: "List and Certificate Complexities in Replicable Learning", Peter Dixon, A. Pavan, Jason Vander Woude, N. V. Vinodchandran "Replicability and stability in learning", Zachary Chase, Shay Moran, Amir Yehudayoff Both of these works study a different notion of replicability that has to do with the number of different outputs of a learning algorithm.
> > "Replicable Reinforcement Learning", Eric Eaton, Marcel Hussing, Michael Kearns, Jessica Sorrell "Replicability in Reinforcement Learning", Amin Karbasi, Grigoris Velegkas, Lin F. Yang, Felix Zhou Both of these papers study replicable algorithms for RL.
> > "Statistical Indistinguishability of Learning Algorithms", Alkis Kalavasis, Amin Karbasi, Shay Moran, Grigoris Velegkas This paper provides a relaxation of the replicability definition and extends some of the equivalences shown in Bun et al. '23 to uncountable domains.
>
> Thank you for pointing them out. We added this to the related work (Section 9).
>
> > Instance-independent regret bound for linear vs. multi-armed bandits: for linear bandits the regret bound is
>  whereas for multi-armed bandits it is. Isn't the former always worse even though there is more structure?
>
> It is always worse in terms of distribution-independent bound because we use a naive bound $N_i^{lin}(p) \le N^{lin}(p)$ for linear bandits, which still outperforms Esfandiari et al. '23.
>
> > Appendix F.2, proof of the (A) case. Isn't there an additive term O(K) missing in the regret bound?
>
> We consider it correct, but let us know if some steps are unclear. We are happy to revise it.
>
> Thank you for many other corrections. We updated the paper accordingly. All changes in revisions are colored in red.

---

> > ### Comment · Reviewer_a2Sd · 2023-11-23
> >
> > I would like to thank the authors for addressing my points and revising their draft accordingly. I have also followed the discussion with Reviewer tkZN. Given the new results that are presented by authors and the updated version of the manuscript, I am increasing my score to 6 since I have a more positive view of the paper.

---

### Official Review · Reviewer_tkZN · 2023-11-04

**Soundness:** 3 good
**Presentation:** 3 good
**Contribution:** 1 poor
**Rating:** 3
**Confidence:** 5

**Summary:**

This paper considers the problem of replicability in stochastic and linear stochastic bandits. Specifically, the $\rho$-replicability criterion requires the sequence of actions chosen by the algorithm in two independent runs over the same data generating process be identical with probability at least $1-\rho$, where $\rho$ is a given replicability parameter, and the probabilities are taken with respect to the internal randomization of the algorithm and randomness in the data generating process. This work considers this criterion in the context of regret minimization in stochastic, and linear stochastic bandits. This paper provides 3 key results: (a) they provide a natural algorithm that achieves an instance dependent expected cumulative regret of $O\left(\rho^{-2}\log T\cdot\min\left(\{\sum_{i\neq i^*} \frac{\Delta_i}{\Delta^2},\sum_{i\neq i^*}\frac{K^2}{\Delta_i}\}\right)\right)$, which, depending on the instance improves upon the existing known regret upper bound of $O(\rho^{-2}\log T\cdot\sum_{i\neq i^*} \frac{K^2}{\Delta_i})$. They additionally provide an instance independent regret upper bound of $O(\rho^{-1}K\sqrt{KT\log T})$ achieved by the same algorithm. (b) They provide a 2-arm regret lower bound of $\Omega(\rho^{-2} \Delta^{-1}\log^{-1}(1/\rho\Delta))$ for any $\rho$-replicable algorithm for stochastic bandits. (c) Lastly, for the case of linear stochastic bandits, they provide an algorithm that builds upon the replicable algorithm for stochastic bandits that achieves an instance dependent regret upper bound of $O(\rho^{-2}\Delta^{-2}d\log T)$, and an instance independent regret upper bound of $O(\rho^{-1}K\sqrt{dKT\log T})$.

**Strengths:**

Overall, the paper is well written and easy to read. The algorithms designed are natural and quite easy to understand.

**Weaknesses:**

To be very honest, I am not too enthusiastic about this result. I have serious questions about the motivation for this particular problem - I don't see the practical need for exact replicability in the bandit setting, as it seems too stringent a condition, and existing ideas (which I shall elaborate shortly) already provide "near"-replicable algorithms for this particular problem at no additional loss in regret. From a technical perspective, I don't see any new technical ideas being developed in this piece of work. The algorithms provided are pretty much a straightforward application of the batched bandits idea of Perchet et. al. (Annals of Statistics, 2016) + its other follow-ups. This is also an obvious connection because the batched bandits framework already provides near-replicability: very arm $i\neq i^*$ essentially has a fixed optimal epoch when the number of plays exceeds $\tilde{O}(\Delta_i^{-2})$ for the first time so it becomes "distinguishable" from the best arm. We have the high probability guarantee that every arm will necessarily be played up until the optimal epoch and will necessarily be discarded at the end of the epoch following the optimal epoch (by slightly strengthening the rejection condition for an arm to be something like the empirical means are separated by 3x the size of the confidence interval in that epoch, and increasing the number of samples obtained from active arms such that the size of the confidence interval shrinks by a factor of 1/5 across successive epochs - this will effectively guarantee that there is only one "uncertain" epoch when an arm may or may not be rejected, before which the arm will necessarily be played and after which the arm will necessarily be rejected. This slight modification of the rejection condition affects cumulative regret by only a small constant). Therefore for every arm, the behavior of the algorithm is near-deterministic and hence replicable (with polynomially high probability in $T$) for all epochs except one epoch where anything can happen (the optimal epoch where this arm becomes distinguishable for the first time. In this epoch, it is hard to argue the nature of the overlap/non-overlap in the confidence intervals since the size of the confidence intervals is at the same scale -- up to small constants -- as the gap between the rewards). One might argue that this is already good enough for most practical purposes - the number of times any arm will be played will be within constant factor of each other across any two runs of the algorithm with any desired polynomially large probability (in $1/T$). In fact, across any two runs of the algorithm, we have the additional guarantee that for any arm, the number of times the arm is played takes only 2 possible values! One can add additional randomness by randomizing the width of the confidence interval in this one epoch (you don't need to know the identity of the bad epoch. you can essentially randomize confidence intervals in each epoch and the strong separation or overlap in confidence intervals in other epochs guarantees that this randomization effectively only kicks in at this one potentially bad epoch where it is hard to argue what the algorithm does) to explicitly get a handle on what the probability of non-replicability looks like. The instance-independent regret bounds follow by an identical analysis (essentially the same analysis as batched bandits), the algorithm and analysis for linear bandits follows pretty much the same way. The lower bound is also extremely weak. These are not at all novel ideas, and I find it hard to argue acceptance for these results.

**Questions:**

What is novel in this work? I would request the authors to improve their upper bounds (which I am almost certain is possible with a finer analysis exploiting subgaussianity and a cleverer randomization strategy for rejection in an epoch), as well as provide tighter lower bounds.

---

> ### Author Response · Authors · 2023-11-17
> **On the relevance to batched bandit problem**
>
> Thank you for your thoughtful comments. The main point is the novelty comparison with the literature by Vianney Perchet, Philippe Rigollet, Sylvain Chassang, Erik Snowberg (2016). "Batched Bandit Problem." Annals of Statistics 2016, Vol. 44, No. 2, 660–681 [PRCS2016].
>
> The batched bandit problem [PRCS2016] utilizes an explore-then-commit policy that conducts uniform exploration for each batch. As you correctly mentioned, if it is applied with geometric size of batches (Sec 4.2 therein), one can obtain the high probability guarantee that the rejection of arms occurs in one of two rounds; we use this fact in our algorithms as well.
> In view of this, our novelty is as follows:
> * The analysis in [PRCS2016] focuses on the case of $2$ arms, whereas our algorithm extends to $K$ arms.
> * We use randomization of the confidence bound so that the reproducibility is guaranteed with probability more than $1-\rho$ for $\rho < 0.5$, unlike the algorithm in [PRCS2016].
> * We formalize the sufficient condition for a sequential algorithm to be replicable, which is implicitly used in [EKKKMV2023]. Thank you for clarifying the multiple correction on replication. In fact, [EKKKMV2023] incorrectly counts the correction factor needed for the multiplity, and the formalization in our Section 3 has a significant contribution.
> * Furthermore, we unify the explore-then-commit [PRCS2016] and successive rejects [GHRZ2019,EKKKMV2023] for $K$ arms in a way that guarantees reproducibility (Algorithm 2).
>
> [GHRZ2019] Zijun Gao, Yanjun Han, Zhimei Ren, Zhengqing Zhou. "Batched Multi-armed Bandits Problem." NeurIPS 2019.
>
> [EKKKMV2023] Hossein Esfandiari and Alkis Kalavasis and Amin Karbasi and Andreas Krause and Vahab Mirrokni and Grigoris Velegkas. "Replicable Bandits." ICLR 2023.
>
> Since these two papers [PRCS2016,GHRZ2019] as well as other papers in batched bandits are clearly relevant to our work, we revised our related work (Section 9) to add a discussion (highlighted in red color). Thank you again for introducing these materials.

---

> > ### Comment · Reviewer_tkZN · 2023-11-22
> > **Still not convinced**
> >
> > Thank you for looking into the batched bandit literature. However, I am still not convinced regarding the novelty in this work. The extension to the $K$ arm case was established in GHRZ2019 following the work of PRCS2016, which is essentially the same as what is done in the current submission. That hardly counts as novelty. I agree that the analysis of batched bandits only guarantees that each arm gets rejected in one of two possible epochs, but I dont believe the leap from that to $1-\rho$ replicability is that novel. As I said earlier, I dont even see a strong motivation for this type of a replicability result, especially given that the batched bandits framework gives "near" replicability anyway (see the comments from my first response). I am really tired of seeing these A+B type results, which are essentially - researchers have historically studied problem A, and in recent years concept B has been developed, so why not study A+B - especially when no new techniques or tools are being developed that would be of broader applicability other than the narrow scope considered in that paper. With regards to this work, at the very least I would have liked to see a tighter lower bound, if not a tighter upper bound, both of which I strongly believe are possible.
> >
> > But that being said, I think novelty is a subjective thing, and although I will not be changing my score, I would like to see what the other reviewer's thoughts are regarding the matter.

---

> ### Author Response · Authors · 2023-11-22
> **Re: Still not convinced**
>
> Dear Reviewer tkZN:
>
> In our understanding, your second comment is summarized as follows:
>
> * You are not convinced of the novelty of the replicable bandit problem setting (K-armed bandits with replicability).
> * You are unsatisfied with the gap between the upper bound and the lower bound.
>
> Let us set the first one aside because it is somewhat subjective as you mentioned. Regarding the second question, you strongly believe you can have a better bound in both lower and upper bound. Please tell us some idea of how the tight bound should looks like? I think you understand the problem structure well, and it would be great if you shared your insight.
>
> Best,
> Authors

---

> ### Comment · Reviewer_tkZN · 2023-11-22
>
> 1. There are two components to my opinion regarding novelty, and both arise from the fact that the batched bandits work is already an established result in this area. (a) Subjective - since batched bandits already provides "near" replicability, I am not convinced about the motivation for $1-\rho$ replicability in the stochastic bandits setting. It is an extremely stringent condition and I cannot think of any practical situation where one might desire these stronger properties over the weaker guarantees already provided by the batching approach which importantly come at no asymptotic loss in regret. (b) Not as subjective - I do not believe there are any new techniques being developed in this work that would be more broadly applicable beyond the scope of the problem considered here, which further limits the impact of this paper. These two factors together make me question the novelty here.
>
> 2.  The point is, there is a big gap between the upper and lower bounds. I find it quite hard to believe that the new replicability lower bound you have introduced must be independent of $T$. If you strongly believe that it must be the case, then that means the correct upper bound you should be aiming toward must contain two terms (additive instead of multiplicative), the first being the usual MAB regret, and the other additive term being the excess regret due to replicability. This seems implausible to me, though I cannot say for certain. I certainly think there is scope for improvement in your upper bounds though, through a more careful analysis exploiting subgaussianity and anti-concentrations to bound the probability of non-replication. That being said, this is as far as I can get based on intuition alone.
>
> Lastly, I do understand that some of my reservations regarding this work are quite subjective. I would like to see both the authors as well as other reviewer's thoughts on the matter.

---

> > ### Author Response · Authors · 2023-11-22
> > **On lower and upper bounds**
> >
> > Thank you for your discussion. We really appreciate that.
> >
> > * Lower bound:
> >
> > Your understanding is quite clear in general. However, on this point, we consider you to be incorrect. The question here is whether the bound is additive
> > $
> > \frac{\log T}{\Delta} + \frac{1}{\Delta \rho^2}
> > $
> > or multiplicative
> > $
> > \frac{\log T}{\Delta \rho^2}.
> > $
> > You argue it should be multiplicative, but we are pretty sure it is additive. This is because
> > * Uniformly goodness [Lai and Robbins 1985] requires us to draw us to the level of confidence $1/T$ at least, so that the expected regret due to event of misidentification $\hat{\mu}_1 < \hat{\mu}_2$ is $o(\log T)$.
> > * $\rho$-replicability requires us to draw of arm $N_2$ less than $T/2$ (say, $1/3$) in model $1$ and more than $T/2$ (say, $2/3$) in model $2$, where the distance of model $1$ and model $2$ must be $O(\rho)$ for replicability, which is what our lower bound states. This has nothing to do with a confidence level of $1/T$, and basically independent.
> >
> > Regarding your next push: Can we achieve an upper bound of additive? We can easily do at some extent. For ease of discussion, we make the union bound on the failure of good event (event $\mathcal{G}$) to be a small level of $O(1/T)$. However, it is okay that this failure is of level $\rho$, then we can have $O(\frac{\log(\log T/\rho)}{\Delta \rho^2} + \frac{\log T}{\Delta})$, where $\log(\log T/\rho)$ comes from union bound of $\log T$ phases of total confidence level $\rho$. Not sure we can shave $\log \log T$ that looks unessential.
> >
> > * Upper bound
> >
> > I see you believe the improvement in upper bound is there but are not sure of the explicit form (or not comfortable for putting it here, maybe... sorry if you feel that). We can propose one. For example, we bound the $O(K)$ regret of Algorithm 1 (which we think novel) in terms $\Delta_K/\Delta_2$, but we can improve it to  $\Delta_K/\Delta_C$ for any constant $C$ w.r.t. $K$, but we do not include it because it is rather nonessential.
> >
> > Let us know if you are unclear about the explanation above. We are happy to elaborate.
> >
> > Best,
> >
> > Authors

---

> ### Comment · Reviewer_tkZN · 2023-11-22
>
> I was not arguing for the lower bound to be multiplicative, though looking back at my comment I understand it came off that way. We are essentially operating with two different "scales" here - one is the replicability parameter $\rho$, and the other is the time horizon $T$, which are independent of each other, due to which the penalty due to replicability will be independent of $T$. I apologize for the carelessness in the writing of my response. The point I was trying to make is that you should be trying to improve your upper bound to $\sum \log T/\Delta_i + 1/(\Delta\rho^2)$, which is substantially stronger than the $O(\sum \Delta_i\log T/\Delta^2)$ that you prove in this paper, which I think follows straightforwardly from existing analyses in the batching space. Moreover, you should at least try to prove a $K$ armed lower bound rather than a 2 armed one. Regarding further improvements to the upper bound, a $\log\log T$ dependence in the term that contains the replicability parameter is something I believed would be possible due to the union bound over only the $\log T$ epochs, but I did not want to commit to it since I have not formally worked out the analysis. If it is truly possible to get a regret that looks like $\sum_i \log\log T/(\rho^2\Delta_i) + \log T/\Delta_i$, it would be a massive improvement over the $O(\sum \Delta_i\log T/(\Delta^2\rho^2))$ regret proved in this work. This would also substantially shrink the gap between your upper and lower bounds, to just a $O(\log\log T)$ factor, which I think is near negligible. I also wonder whether its possible to achieve an even stronger result for worst-case regret (additive, with a $\log\log\log T$ dependence in the term that contains the replicability parameter $\rho$), since you can achieve near-optimal $O(\sqrt{KT\log T})$ regret in just $\log\log T$ batches in the usual non-replicable case, as opposed to $\log T$ batches required to achieve the optimal $O(\sum_i \log T/\Delta_i)$ instance-dependent regret. If this is indeed the case, I would indeed be willing to change my evaluation of this work. I still have my reservations about the motivation behind this setting, and that even this further improvement comes from a more careful analysis of existing batching ideas rather than the introduction of truly new tools, the fact that this work would then provide a near-complete resolution (up to low order terms) to the replicability question would make up for these deficiencies.

---

> ### Author Response · Authors · 2023-11-22
> **What we can further do easily and challenging**
>
> Thank you for your reply.
>
> * We think we can extend our lower bound from $2$ to $K$-arms relatively easily to get $\sum_i (\frac{\log T}{\Delta} + \frac{1}{\Delta \rho^2})$ bound that you consider ideal.
> * We hypothesize (though we are not very sure) that it is impossible to have this upper bound of $\sum_i (\frac{\log T}{\Delta} + \frac{1}{\Delta \rho^2})$ because it implies $K-1$ decisions of eliminating each suboptimal arm at some round, and it is subject to $O(K)$ different decision points. We guess there are some limitations (Algorithm 1 focused on deleting all arms at the same time to the cost of $\Delta_K/\Delta_2$ coefficient as you correctly pointed out). But we are not sure; our current framework is just incapable of removing this $K$ multiplication.
> * Worst-case bound is easily improvable to replace $\sqrt{\log T}/\rho$ of the current paper with $\sqrt{\log \log T}\rho + \sqrt{\log T}$. Moreover, if we can reduce the number of batches to $\log \log T$ as you suggested, then we can further improve upon it. This was not in our mind, thank you for your suggestion.

---

> ### Comment · Reviewer_tkZN · 2023-11-22
>
> For worst-case regret, you can be a lot more aggressive with batch sizes - using a squaring trick to get $\log\log T$ batches/adaptive rounds in total achieving $\tilde{O}(\sqrt{KT})$ regret rather than the instance-dependent case which requires a more conservative doubling trick to get $\log T$ batches/adaptive rounds in total achieving $\sum \log T/\Delta_i$ regret.
>
> In any case, I think we have made all our points, and it is up to other reviewers and the meta reviewers to weigh in with their opinions.

---

### Author Response · Authors · 2023-11-23
**Thank you for discussion**

We thank the reviewers for many discussions.

* In accordance with the discussion with Rev tkZN, we will surely change the amount of exploration so that the confidence level of good event $\mathcal{G}$ is $\rho$ rather than $1/T$. This will replace all upper bounds of $\log T/\rho^2$ in Table 1 with $\log(\log T/\rho)/\rho^2 + \log T$. In other words, the uniformly good term of $\log T$ and replicability term of $1/\rho^2$ are almost separated, which aligns with our lower bound.

We will try to implement other improvements that were raised during the discussion.

---

### Meta-Review · Area_Chair_gr7b · 2023-12-09

**Metareview:**

This paper studies _replicable bandit algorithms_, whose decisions are (w.h.p.) reproducible. Using a definition of replicability due to Esfandiari et al. (2023), a bandit algorithm is replicable if, with probability $\geq 1 - \rho$ over the environment's randomness, it pulls the exact same sequence of arms when executed twice in the same environment. The main contributions are:
* In the $K$-armed (non-contextual) bandit setting, an instance-_dependent_ bound of $O(K)$, and an instance-_independent_ bound of $O(K^{1.5})$. These are improvements over Esfandiari et al.'s bounds, which are $O(K^3)$.
* Lower bounds that show the dependence on $\rho$ is tight.
* For linear bandit problems in $d$ dimensions, an instance-_dependent_ bound of $O(d)$ and an instance-_independent_ bound of $O(K \sqrt{K d})$. Again, these are improvements over Esfandiari et al.'s bounds, which are $O(K^2)$.
* An empirical studies demonstrates that the proposed algorithms work.

The reviewers agreed that the paper is well written and easy to follow (except for the proofs). Further, except for an issue with one proof (caught by reviewer `apnD` and swiftly corrected by the authors), the work is sound. The bounds appear to be the tightest yet for the problem of replicable bandits.

There was some question as to the novelty or magnitude of the contributions, given that they rely heavily on Esfandiari et al. and Impagliazzo et al. (2022) and Perchet et al. (2016). This may just be an issue of writing, clarifying the delta from prior work. Relatedly, claims that the proposed algorithms are significantly "simpler" than Esfandiari et al.'s may be inflated (see reviewer `a2Sd `).

Reviewer `tkZN` questioned whether the replicability definition itself is overly strict and, hence, impractical. They claimed that an adaptation of a batched bandit algorithm can achieve "near-replicability" with similar regret guarantees, thereby obviating the current work. They also felt that the gap between the upper and lower regret bounds was too wide. In spite of many rounds of interaction with the authors, these concerns were not resolved. Reviewer `Hx4P` was similarly unconvinced. They also felt that the connection to differentially private bandits should be made.

In contrast, the author-reviewer discussion period prompted reviewers `a2Sd` and `apnD` to raise their overall scores to 6, leaning toward acceptance.

This paper is currently on the border, and I am unfortunately leaning toward "reject." I encourage the authors to incorporate the feedback from the reviews into the next version of the paper.

**Justification For Why Not Higher Score:**

Reviewer `tkZN` raises a legitimate concern. Even if the proposed method is ultimately better than `tkZN`'s suggestion, this comparison should be addressed in a future version of the paper. This type of thing shouldn't be left for the camera-ready.

**Justification For Why Not Lower Score:**

N/A

---

### Decision · Program_Chairs · 2024-01-16

Reject